# Bubblewrap: Online tiling and real-time flow prediction on neural manifolds

**Anne Draelos**
Biostatistics & Bioinformatics
Duke University
anne.draelos@duke.edu

**Pranjal Gupta**
Psychology & Neuroscience
Duke University
pranjal.gupta@duke.edu

**Na Young Jun**
Neurobiology
Duke University
nayoung.jun@duke.edu

**Chaichontat Sriworarat**
Biomedical Engineering
Duke University
chaichontat.s@duke.edu

**John Pearson**
Biostatistics & Bioinformatics
Electrical & Computer Engineering
Neurobiology
Psychology & Neuroscience
Duke University
john.pearson@duke.edu

## Abstract

While most classic studies of function in experimental neuroscience have focused on the coding properties of individual neurons, recent developments in recording technologies have resulted in an increasing emphasis on the dynamics of neural populations. This has given rise to a wide variety of models for analyzing population activity in relation to experimental variables, but direct testing of many neural population hypotheses requires intervening in the system based on current neural state, necessitating models capable of inferring neural state online. Existing approaches, primarily based on dynamical systems, require strong parametric assumptions that are easily violated in the noise-dominated regime and do not scale well to the thousands of data channels in modern experiments. To address this problem, we propose a method that combines fast, stable dimensionality reduction with a soft tiling of the resulting neural manifold, allowing dynamics to be approximated as a probability flow between tiles. This method can be fit efficiently using online expectation maximization, scales to tens of thousands of tiles, and outperforms existing methods when dynamics are noise-dominated or feature multi-modal transition probabilities. The resulting model can be trained at kiloHertz data rates, produces accurate approximations of neural dynamics within minutes, and generates predictions on submillisecond time scales. It retains predictive performance throughout many time steps into the future and is fast enough to serve as a component of closed-loop causal experiments.

## 1 Introduction

Systems neuroscience is in the midst of a data explosion. Advances in microscopy [1, 2] and probe technology [3, 4, 5] have made it possible to record thousands of neurons simultaneously in behaving

35th Conference on Neural Information Processing Systems (NeurIPS 2021).

animals. At the same time, growing interest in naturalistic behaviors has increased both the volume and complexity of jointly recorded behavioral data. On the neural side, this has resulted in a host of new modeling and analysis approaches that aim to match the complexity of these data, typically using artificial neural network models as proxies for biological neural computation [6, 7, 8].

At the same time, this increase in data volume has resulted in increasing emphasis on methods for dimensionality reduction [9] and a focus on neural populations in preference to the coding properties of individual neurons [10]. However, given the complexity of neural dynamics, it remains difficult to anticipate what experimental conditions will be needed to test population hypotheses in *post hoc* analyses, complicating experimental design and reducing power. Conversely, adaptive experiments, those in which the conditions tested change in response to incoming data, have been used in neuroscience to optimize stimuli for experimental testing [11, 12, 13, 14], in closed-loop designs [15, 16, 17], and even to scale up holographic photostimulation for inferring functional connectivity in large circuits [18].

Yet, despite their promise, adaptive methods are rarely applied in practice for two reasons: First, although efficient online methods for dimensionality reduction exist [19, 20, 21, 22, 23], these methods do not typically identify *stable* dimensions to allow low-dimensional representations of data to be compared across time points. That is, when the spectral properties of the data are changing in time, methods like incremental SVD may be projecting the data into an unstable basis, rendering these projections unsuitable as inputs to further modeling. Second, while many predictive models based on the dynamical systems approach exist [6, 24, 25, 26, 27, 28, 29], including online approaches [30, 31, 16, 32], they typically assume a system with lawful dynamics perturbed by Gaussian noise. However, many neural systems of interest are noise-dominated, with multimodal transition kernels between states.

In this work, we are specifically interested in closed loop experiments in which predictions of future neural state are needed in order to time and trigger interventions like optogenetic stimulation or a change in visual stimulus. Thus, our focus is on predictive accuracy, preferably far enough into the future to compensate for feedback latencies. To address these goals, we propose an alternative to the linear systems approach that combines a fast, stable, online dimensionality reduction with a semiparametric tiling of the low-dimensional neural manifold. This tiling introduces a discretization of the neural state space that allows dynamics to be modeled as a Hidden Markov Model defined by a sparse transition graph. The entire model, which we call "Bubblewrap," can be learned online using a simple EM algorithm and handles tilings and graphs of up to thousands of nodes at kiloHertz data rates. Most importantly, this model outperforms methods based on dynamical systems in high-noise regimes when the dynamics are more diffusion-like. Training can be performed at a low, fixed latency $\approx$10ms using a GPU, while a cached copy of the model in main memory is capable of predicting upcoming states at <1ms latency. As a result, Bubblewrap offers a method performant and flexible enough to serve as a neural prediction engine for causal feedback experiments.

## 2    Stable subspaces from streaming SVD

As detailed above, one of the most pressing issues in online neural modeling is dealing with the increasingly large dimensionality of collected data — hundreds of channels per Neuropixels probe [4, 5], tens of thousands of pixels for calcium imaging. However, as theoretical work has shown [33, 34], true neural dynamics often lie on a low-dimensional manifold, so that population activity can be accurately captured by analyzing only a few variables.

Here, we combine two approaches to data reduction: In the first stage, we use sparse random projections to reduce dimensionality from an initial $d$ dimensions (thousands) to $n$ (a few hundred) [35, 36]. By simple scaling, for a fixed budget of $N$ cells in our manifold tiling, we expect density (and thus predictive accuracy) to scale as $N^{\frac{1}{n}}$ in dimension $n$, and so we desire $n$ to be as small as possible. However, by the Johnson-Lindenstrauss Lemma [37, 36], when reducing from $d$ to $n$ dimensions, the distance between vectors $u_*$ and $v_*$ in the reduced space is related to the distance between their original versions $u$ and $v$ by

$$(1 - \varepsilon)\|u - v\|^2 \leq \|u_* - v_*\|^2 \leq (1 + \varepsilon)\|u - v\|^2 \tag{1}$$

with probability $1 - \delta$ if $n > \mathcal{O}(\log(1/\delta)/\varepsilon^2)$. Unfortunately, even for $\varepsilon \sim 0.1$ (10% relative error), the required $n$ may be quite large, making this inappropriate for reducing to the very small numbers of effective dimensions characterizing neural datasets.

**Algorithm 1** Procrustean SVD (proSVD)

---

1: **Given:** Initial data $X_0$, decay parameter $\alpha \in (0, 1]$
2: **Initialize:** QR Factorization: $X_0 = Q_0 R_0$
3:
4: **for** $t = 1 \ldots$ **do**
5:      Fetch $b$ new columns of data, $X_+$
6:      $C \leftarrow Q_{t-1}^\top X_+, \quad X_\perp \leftarrow X_+ - Q_{t-1}C, \quad Q_\perp, R_\perp \leftarrow \texttt{QR}(X_\perp)$          ▷ Gram-Schmidt
7:      $\hat{Q} \leftarrow [Q_{t-1} \quad Q_\perp], \quad \hat{R} \leftarrow \begin{bmatrix} R_{t-1} & C \\ 0 & R_\perp \end{bmatrix}$          ▷ QR of augmented data
8:      $U, \Sigma, V \leftarrow \texttt{SVD}(\hat{R})$
9:      $\Sigma \leftarrow \alpha\Sigma$          ▷ Discount old data
10:      $M \leftarrow Q_{t-1}^\top \hat{Q} U_1 = [\mathbb{1}_{k \times k} \quad \mathbf{0}_{k \times b}] U_1$          ▷ $U_1$ contains the first $k$ columns of $U$
11:      $\tilde{U}, \tilde{\Sigma}, \tilde{V} \leftarrow \texttt{SVD}(M), \quad T \leftarrow \tilde{U}\tilde{V}^\top$      ▷ Orthogonal Procrustes: $\min_T \|\hat{Q}U_1 T^\top - Q_{t-1}\|_F$
12:      $Q_t \leftarrow \hat{Q}U_1 T^\top$          ▷ Update left subspace basis
13:      $Q_v, R_v \leftarrow \texttt{QR}(V), \quad R_t \leftarrow T\Sigma_1 Q_v^\top$          ▷ QR right subspace, update inner block
14: **end for**

---

Thus, in the second stage, we reduce from $n \sim \mathcal{O}(100)$ to $k \sim \mathcal{O}(10)$ dimensions using a streaming singular value decomposition. This method is based on the incremental block update method of [20, 22] with an important difference: While the block update method aims to return the top-$k$ SVD at every time point, the directions of the singular vectors may be quite variable during the course of an experiment (Figure 1d–h), which implies an unstable representation of the neural manifold. However, as we show below, the top-$k$ *subspace* spanned by these vectors stabilizes in seconds on typical neural datasets and remains so throughout the experiment. Therefore, by selecting a stable basis (instead of the singular vector basis) for the top-$k$ subspace, we preserve the same information while ensuring a stable representation of the data for subsequent model fitting.

More specifically, let $\mathbf{x}_t \in \mathbb{R}^n$ be a vector of input data after random projections. In our streaming setup, these are processed $b$ samples at a time, with $b = 1$ reasonable for slower methods like calcium imaging and $b = 40$ more appropriate for electrophysiological sampling rates of ~20kHz. Then, if the data matrix $X$ has dimension $n \times T$, adding columns over time, the incremental method of [20, 22] produces at each time step a factorization $X = QRW^\top$, where the columns of the orthogonal matrices $Q$ and $W$ span the left and right top-$k$ singular subspaces, respectively. If the matrix $R$ were diagonal, this would be equivalent to the SVD. In the incremental algorithm, $R$ is augmented at each timestep based on new data to form $\hat{R}$, which is block diagonalized via an orthogonal matrix and truncated to the top-$k$ subspace, allowing for an exact reduced-rank SVD (Appendix A).

However, as reviewed in [20, 22], since there are multiple choices of basis $Q$ for for the top-$k$ singular subspace, there are likewise multiple choices of block diagonalization for $\hat{R}$. In [20, 22], the form of this operation is chosen for computational efficiency. But an equally valid option is to select the orthogonal matrix that minimizes the change in the singular subspace basis $Q$ from one timestep to the next:

$$\min\|Q_t - Q_{t-1}\|_F = \min_T \|\hat{Q}U_1 T^\top - Q_{t-1}\|_F, \tag{2}$$

where $\hat{Q}$ is an augmented basis for the top-$(k + b)$ singular subspace, $U_1$ contains the first $k$ left singular vectors of $\hat{R}$, and $T$ is an orthogonal matrix (Appendix A). This minimization is known as the Orthogonal Procrustes problem and has a well-known solution [38]: $T = \tilde{U}\tilde{V}^\top$, where $\tilde{U}$ and $\tilde{V}$ are the left and right singular vectors, respectively, of $M \equiv Q_{t-1}^\top \hat{Q}U_1$. (See [39] for a recent application of similar ideas in brain-computer interfaces). This Procrustean SVD (proSVD) procedure is summarized in Algorithm 1. There, lines 1–8 follow [20, 22], while lines 10 and 11 perform the Orthogonal Procrustes procedure. Line 9 serves as a leak term that discounts past data as in [40].

Figures 1a-c illustrates the performance of the two-stage dimension reduction algorithm for a case of $d = 10^4$ randomly generated Gaussian data. While proSVD yields minimal distortion (due to truncation of the spectrum to $k = 6$), random projections require $k \sim \mathcal{O}(100)$ to achieve the same result (Figure 1a). By contrast, random projections are much faster (Figure 1b). Thus, we can trade off distortion against time by adjusting $n$, the number of intermediate dimensions. As Figure 1c shows, the optimal tradeoff occurs somewhere around $n = 200$ for this example.

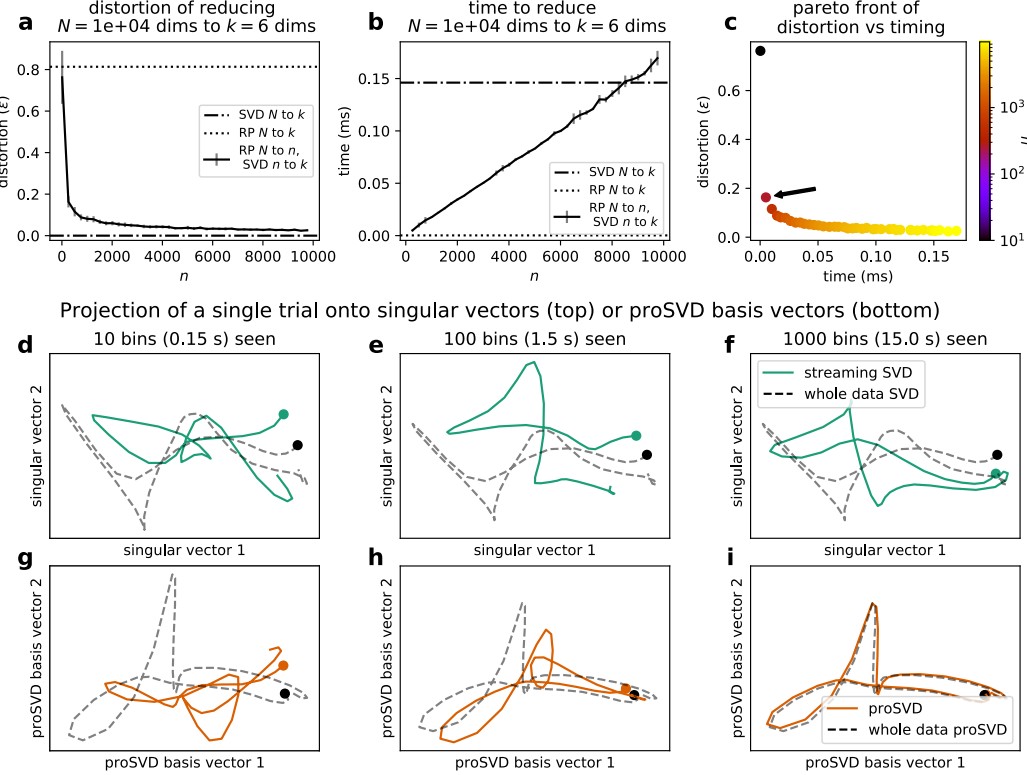

Figure 1: **Timing and stability of two-stage dimension reduction. a)** Distortion ($\varepsilon$) as a function of number of dimensions retained ($n$) for both sparse random projections and proSVD on random Gaussian data with batch size $b = 1000$. **b)** Time required for the dimensionality reduction in **(a)**, amortized for batch size. While random projections are extremely efficient, proSVD time costs grow with the number of dimensions retained. **c)** Pareto front for the time-distortion tradeoff of random projections followed by proSVD. Color indicates $n$, the number of dimensions retained by random projections. Black arrow indicates the particular tradeoff we chose of $n = 200$. **d–f)** Embedding of a single trial (green line) into the basis defined by streaming SVD for different amounts of data seen. Dotted line indicates the same trial embedded using SVD on the full data set. Rapid changes in estimates of singular vectors early on lead to an unstable representation. **g–i)** Same trial and conventions as **(d–f)** for the proSVD embedding. Dotted lines in the two rows are the same curve in different projections.

Figures 1d-i show results for neural data from recorded from monkey motor cortex [26] in a cued reach task. While projection of the data into the basis defined by streaming SVD remains unstable early in data collection (top), the proSVD representation is nearly equivalent to the full offline result after only a few trials ($\approx$15s, middle). This is due to the fact that, in all data sets we examined, the top-$k$ SVD *subspace* was identified extremely quickly; proSVD simply ensures the choice of a stable basis for that subspace.

## 3  Bubblewrap: a soft manifold tiling for online modeling

As reviewed above, most neural population modeling approaches are based on the dynamical systems framework, assuming a lawful equation of motion corrupted by noise. However, for animals engaged in task-free natural behavior [41, 42, 43], trajectories are likely to be sufficiently complex that simple dynamical models fail. For instance, dynamical systems models with Gaussian noise necessarily produce unimodal transition probabilities centered around the mean prediction, while neural trajectories may exhibit multimodal distributions beginning at the same system state. By contrast, we pursue an

alternative method that trades some accuracy in estimating instantaneous system state for flexibility in modeling the manifold describing neural activity.

Our approach is to produce a soft tiling of the neural manifold in the form of a Gaussian mixture model (GMM), each component of which corresponds to a single tile. We then approximate the transitions between tiles via a Hidden Markov Model (HMM), which allows us to capture multimodal probability flows. As the number of tiles increases, the model produces an increasingly finer-grained description of dynamics that assumes neither an underlying dynamical system nor a particular distribution of noise.

More specifically, let $x_t$ be the low-dimensional system state and let $z_t \in 1 \dots N$ index the tile to which the system is assigned at time $t$. Then we have for the dynamics

$$p(z_t = j | z_{t-1} = i) = A_{ij} \quad p(x_t | z_t) = \mathcal{N}(\mu_{z_t}, \Sigma_{z_t}) \quad p(\mu_j, \Sigma_j) = \text{NIW}(\mu_{0j}, \lambda_j, \Psi_j, \nu_j), \quad (3)$$

where we have assumed Normal-inverse-Wishart priors on the parameters of the Gaussians. Given its exponential family form and the conjugacy of the priors, online expectation maximization updates are available in closed form [44, 45, 46] for each new datum, though we opt, as in [45] for a gradient-based optimization of an estimate of the evidence lower bound

$$\mathcal{L}(A, \mu, \Sigma) = \sum_{ij} (\hat{N}_{ij}(T) + \beta_{ij} - 1) \log A_{ij} + \sum_j (\hat{S}_{1j}(T) + \lambda_j \mu_{0j})^\top \Sigma_j^{-1} \mu_j \qquad (4)$$

$$- \frac{1}{2} \sum_j \text{tr}((\Psi_j + \hat{S}_{2j}(T) + \lambda_j \mu_{0j} \mu_{0j}^\top + (\lambda_j + \hat{n}_j(T)) \mu_j \mu_j^\top) \Sigma_j^{-1})$$

$$- \frac{1}{2} \sum_j (\nu_j + \hat{n}_j(T) + d + 2) \log \det \Sigma_j$$

with accumulating (estimated) sufficient statistics

$$\alpha_j(t) = \sum_i \alpha_i(t-1) \Gamma_{ij}(t) \qquad \hat{N}_{ij}(t) = (1 - \varepsilon_t) \hat{N}_{ij}(t-1) + \alpha_i(t-1) \Gamma_{ij}(t) \qquad (5)$$

$$\hat{n}_j(t) = \sum_i \hat{N}_{ij}(t) \qquad \hat{S}_{1j}(t) = (1 - \varepsilon_t) \hat{S}_{1j}(t-1) + \alpha_j(t) x_t$$

$$\hat{S}_{2j}(t) = (1 - \varepsilon_t) \hat{S}_{2j}(t-1) + \alpha_j(t) x_t x_t^\top$$

where $\alpha_j(t) = p(z_t = j | x_{1:t})$ is the filtered posterior, $\Gamma_{ij}(t)$ is the update matrix from the forward algorithm [44], and $\varepsilon_t$ is a forgetting term that discounts previous data. Note that even for $\varepsilon = 0$, $\mathcal{L}$ is only an estimate of the evidence lower bound because the sufficient statistics are calculated using $\alpha(t)$ and not the posterior over all observed data.

In setting Normal-Inverse-Wishart priors over the Gaussian mixture components, we take an empirical Bayes approach by setting prior means $\mu_{0j}$ to the current estimate of the data center of mass and prior covariance parameters $\Psi_j$ to $N^{-\frac{2}{k}}$ times the current estimate of the data covariance (Appendix B). For initializing the model we use a small data buffer $M \sim \mathcal{O}(10)$. We chose effective observation numbers $(\lambda, \nu) = 10^{-3}$ and trained this model to maximize $\mathcal{L}(A, \mu, \Sigma)$ using Adam [47], enforcing parameter constraints by replacing them with unconstrained variables $a_{ij}$ and lower triangular $L_j$ with positive diagonal: $A_{ij} = \exp(a_{ij}) / \sum_j \exp(a_{ij})$, $\Sigma_j^{-1} = L_j L_j^\top$.

Finally, in order to prevent the model from becoming stuck in local minima and to encourage more effective tilings, we implemented two additional heuristics as part of Bubblewrap: First, whenever a new observation was highly unlikely to be in any existing mixture component ($\log p(x_t | z_t) < \theta_n$ for all $z_t$), we teleported a node at this data point by assigning $\alpha_J(t) = 1$ for an unused index $J$. For initial learning this results in a "breadcrumbing" approach where nodes are placed at the locations of each new observed datum. Second, when the number of active nodes was equal to our total node budget $N$, we chose to reclaim the node with the lowest value of $\hat{n}(t)$ and zeroed out its existing sufficient statistics before teleporting it to a new location. In practice, these heuristics substantially improved performance, especially early in training (Appendix D). The full algorithm is summarized in Algorithm 2.

**Algorithm 2** Bubblewrap

1: **Given:** Hyperparameters $\lambda_j, \nu_j, \beta_t$, forgetting rate $\varepsilon_t$, teleport threshold $\theta$, step size $\delta$, initial data buffer $M$
2: **Initialize** with $\{x_1 \ldots x_M\}$: $\mu_j \leftarrow \bar{\mu}, \Sigma_j \leftarrow \bar{\Sigma}, a_{ij} \leftarrow \frac{1}{N}, \alpha_j \leftarrow \lambda_j \frac{1}{N}$.
3:
4: **for** $t = 1 \ldots$ **do**
5:     Observe new data point $x_t$.
6:     **if** $\log p(x_t|z_t) < \theta \; \forall z_t$ **then**                                      ▷ Teleport
7:         $\mu_J = x_t, \alpha_J(t) = 1$ for $J = \arg\min_j \hat{n}_j(t)$
8:     **end if**
9:     Calculate $\Gamma_{ij}(t)$ via forward filtering [44].
10:     Update sufficient statistics via (5).                                     ▷ E step
11:     $\bar{\mu} \leftarrow \frac{\sum_j \hat{S}_{1j}}{\sum_j \hat{n}_j}, \overline{\Sigma} \leftarrow \frac{\sum_j \hat{S}_{2j}}{\sum_j \hat{n}_j} - \bar{\mu}\bar{\mu}^T$         ▷ Global mean and covariance update
12:     $\epsilon_j \sim \mathcal{N}(0, \eta^2), \mu_{0j} \leftarrow a\mu_{0j} + (1-a)\bar{\mu} + \epsilon_j, \Psi_j \leftarrow \frac{\overline{\Sigma}}{N^{\frac{2}{k}}}$     ▷ Update priors (Appendix B)
13:     Perform gradient-based update of $\mathcal{L}(A, \mu, \Sigma)$ (4)                        ▷ M step
14: **end for**

## 4 Experiments

We demonstrated the performance of Bubblewrap on both simulated non-linear dynamical systems and experimental neural data. We compared these results to two existing online learning models for neural data, both of which are based on dynamical systems [30, 32]. To simulate low-dimensional systems, we generated noisy trajectories from a two-dimensional Van der Pol oscillator and a three-dimensional Lorenz attractor. For experimental data, we used four publicly available datasets from a range of applications: 1) trial-based spiking data recorded from primary motor cortex in monkeys performing a reach task [48, 49] preprocessed by performing online jPCA [49]; 2) continuous video data and 3) trial-based wide-field calcium imaging from a rodent decision-making task [50, 51]; 4) high-throughput Neuropixels data [52, 53].

For each data set, we gave each model the same data as reduced by random projections and proSVD. For comparisons across models, we quantified overall model performance by taking the mean log predictive probability over the last half of each data set (Table 1). For Bubblewrap, prediction $T$ steps into the future gives

$$\log p(x_{t+T}|x_{1:t}) = \log \sum_{i,j} p(x_{t+T}|z_{t+T} = j)p(z_{t+T} = j|z_t = i)p(z_t = i|x_{1:t})$$

$$= \log \sum_{i,j} \mathcal{N}(x_{t+1}; \mu_j, \Sigma_j)(A^T)_{ij}\alpha_i(t), \tag{6}$$

where $A^T$ is the $T$-th power of the transition matrix. Conveniently, these forward predictions can be efficiently computed due to the closed form (6), while similar predictions in comparison models [30, 32] must be approximated by sampling (Appendix C). In addition, for Bubblewrap, which is focused on coarser transitions between tiles, we also report the entropy of predicted transitions:

$$H(t, T) = -\sum_j p(z_{t+T} = j|x_{1:t}) \log p(z_{t+T} = j|x_{1:t}) = -\sum_{ij} (A^T)_{ij}\alpha_i(t) \log \sum_k (A^T)_{kj}\alpha_k(t). \tag{7}$$

Additional detailed experimental results and benchmarking of our GPU implementation in JAX [54] are in Appendix D. We compared performance of our algorithm against both [30] (using our own implementation in JAX) and Variational Joint Filtering [32] (using the authors' implementation). Our implementation of Bubblewrap, as well as code to reproduce our experiments, is open-source and available online at `http://github.com/pearsonlab/Bubblewrap`.

When tested on low-dimensional dynamical systems, Bubblewrap successfully learned tilings of both neural manifolds, outperforming VJF [32] on both datasets (Figure 2a,b) while it was comparable to the algorithm of [30] on one of the 2D (but neither of the 3D) cases (Figure 2). This is surprising, since both comparison methods assume an underlying dynamical system and attempt to predict

Table 1: Model comparison results as mean $\pm$ standard deviation of the log predictive probability over the last half of the dataset. Asterisks (*) indicate models that degenerated to a random walk.

| | Log predictive probability | | |
|---|---|---|---|
| Dataset | Bubblewrap | VJF [32] | ZP (2016) [30] |
| 2D Van der Pol, 0.05 | $0.965 \pm 1.123$ | $-0.338 \pm 0.427$ | $0.121 \pm 0.857$ |
| 2D Van der Pol, 0.20 | $-1.088 \pm 1.184$ | $-1.140 \pm 0.879$ | $-0.506 \pm 0.964$ |
| 3D Lorenz, 0.05 | $-7.338 \pm 1.289$ | $-16.98 \pm 1.923$ | $-12.39 \pm 1.723*$ |
| 3D Lorenz, 0.20 | $-7.474 \pm 1.279$ | $-17.30 \pm 2.112$ | $-12.42 \pm 1.708*$ |
| Monkey reach | $3.046 \pm 4.959$ | $-5.159 \pm 0.987$ | $3.818 \pm 9.118$ |
| Wide-field calcium | $5.974 \pm 2.979$ | $3.768 \pm 6.204$ | $1.613 \pm 4.083$ |
| Mouse video | $-10.93 \pm 2.386$ | $-15.86 \pm 1.084$ | $-10.65 \pm 4.145*$ |
| Neuropixels | $-12.84 \pm 6.017$ | $-12.06 \pm 5.244$ | $-12.28 \pm 4.567$ |

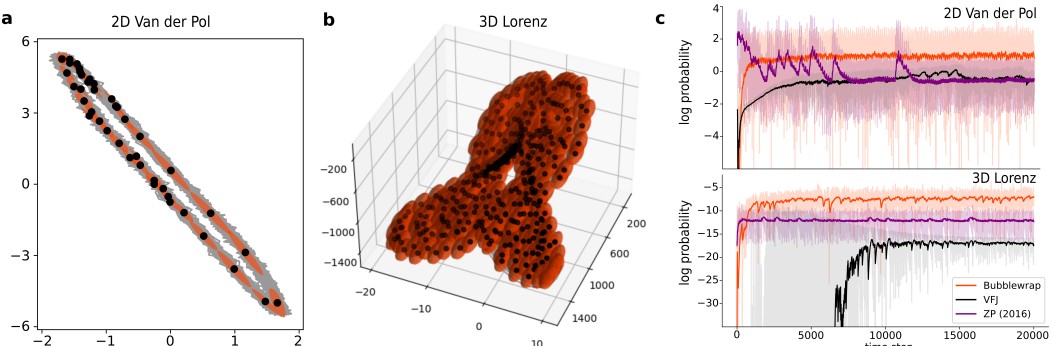

Figure 2: **Modeling of low-dimensional dynamical systems. a)** Bubblewrap end tiling of a 2D Van der Pol oscillator (data in gray; 5% noise case corresponding to line 1 of Table 1). Tile center locations are in black with covariance 'bubbles' for 3 sigma in orange. **b)** Bubblewrap end tiling of a 3D Lorenz attractor (5% noise), where tiles are plotted similarly to (a). **c)** Log predictive probability across all timepoints for each comparative model for the 2D Van der Pol, 0.05 case (top) and for the 3D Lorenz, 0.05 case (bottom).

differences between individual data points, while Bubblewrap only attempts to localize data to within a coarse area of the manifold.

We next tested each algorithm on more complex data collected from neuroscience experiments. These data exhibited a variety of structure, from organized rotations (Figure 3a) to rapid transitions between noise clusters (Figure 3b) to slow dynamics (Figure 3c). In each case, Bubblewrap learned a tiling of the data that allowed it to equal or outperform state predictions from the comparison algorithms (Figure 3d–f, blue). In some cases, as with the mouse dataset, the algorithm of [30] produced predictions for $x_t$ by degenerating to a random walk model (Table 1 marked with *; Appendix D). Regardless, Bubblewrap's tiling generated transition predictions with entropies far below those of a random walk (Figure 3d–f, green), indicating it successfully identified coarse structure, even in challenging datasets. Thus, even though these data are noise-dominated and lack much of the typical structure identified by neural population models, coarse-graining identifies some reliable patterns.

We additionally considered the capability of our algorithm to scale to high-dimensional or high-sampling rate data. As a case study, we considered real-time processing (including random projections, proSVD, and Bubblewrap learning) of Neuropixels data comprising 2688 units with 74,330 timepoints from 30 ms bins. As Figure 4 shows, Bubblewrap once again learns a tiling of the data manifold (a), capturing structure in the probability flow within the space (b) with predictive performance comparable to finer-grained methods (Table 1). More importantly, all these steps can be performed well within the 30ms per sample time of the data (c). In fact, when testing on representative examples of $d = 10^4$ dimensions, 1 kHz sampling rates, or $N = 20,000$ tiles, our algorithm was able to maintain amortized per-sample processing times below those of data acquisition. In practice, we

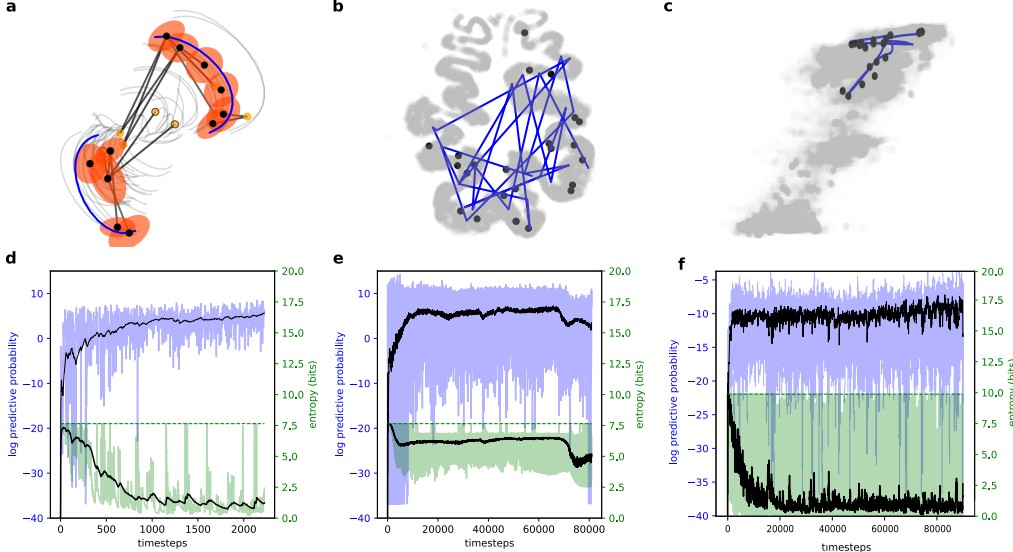

Figure 3: **Bubblewrap results on experimental datasets. a)** Bubblewrap results for example trials (blue) from the monkey reach dataset [48, 49], projected onto the first jPCA plane. All trials are shown in gray. The tile center locations which were closest to the trajectories are plotted along with their covariance "bubbles." Additionally, large transition probabilities from each tile center are plotted as black lines connecting the nodes. Bubblewrap learns both within-trial and across-trial transitions, as shown by the probability weights. **b)** Bubblewrap results on widefield calcium imaging from [50, 51], visualized with UMAP. A single trajectory comprising ≈1.5s of data is shown in blue. Covariance "bubbles" and transition probabilities omitted for clarity. **c)** Bubblewrap results when applied to videos of mouse behavior [50, 51], visualized by projection onto the first SVD plane. Blue line: 3.3s of data. **d, e, f)** Log predictive probability (blue) and entropy (green) over time for the respective datasets in (a,b,c). Black lines are exponential weighted moving averages of the data. Dashed green line indicates maximum entropy ($\log_2(N)$).

found that even in higher-dimensional datasets (as in the Neuropixels case), only 1-2 thousand tiles were used by the model, making it easy to run at kHz data rates. What's more, while learning involved round trip GPU latencies to perform gradient updates, online predictions using slightly stale estimates of Bubblewrap parameters could be performed far faster, in tens of microseconds.

Just as importantly, when used for closed loop experiments, algorithms must be able to produce predictions far enough into the future for interventions to be feasible. Thus we examined the performance of our algorithm and comparison models for predicting $T$ steps ahead into the future. Bubblewrap allows us to efficiently calculate predictions even many time steps into the future using (6), whereas the comparison models require much costlier sampling approaches. Figure 5 shows the mean log predictive probabilities for all models many steps into the future for each experimental dataset (top row), and the entropy of the predicted transitions using Bubblewrap (bottow row). Our algorithm consistently maintains performance even when predicting 10 steps ahead, providing crucial lead time to enable interventions at specific points along the learned trajectory. In comparison, predictive performance of [30], which initially matches or exceeds Bubblewrap for two datasets, rapidly declines, while Variational Joint Filtering [32], with lower log likelihood, also exhibits a slow decay in accuracy.

## 5 Discussion

While increasing attention has been paid in neuroscience to population hypotheses of neural function [10], and while many methods for modeling these data offline exist, surprisingly few methods function online, though presumably online methods will be needed to test some population dynamics hypotheses [17]. While the neural engineering literature has long used online methods based on

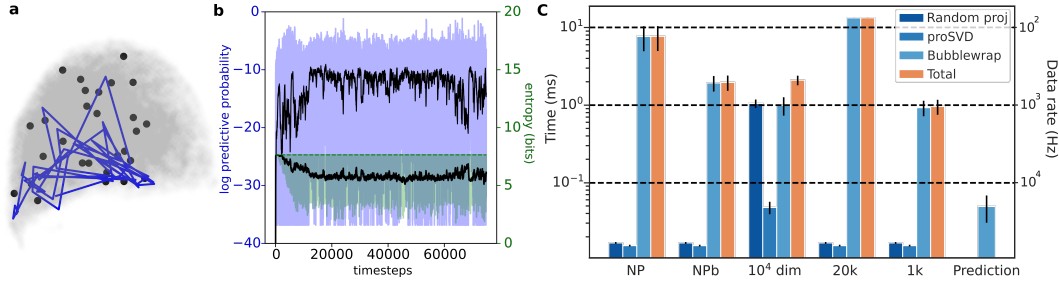

Figure 4: **High-throughput data & benchmarking. a)** Bubblewrap results for example trajectories (blue) in the Neuropixels dataset [52, 53] (data in gray) visualized with UMAP. **b)** Log predictive probability (blue) and entropy (green) over time. Black lines are exponential weighted moving averages of the data. Dashed green line indicates maximum entropy. **c)** Average cycle time (log scale) during learning or prediction (last bar) for each timepoint. Neuropixels (NP) is run as in (a,b) with no optimization and all heuristics, and Bubblewrap is easily able to learn at rates much faster than acquisition (30 ms). By turning off the global mean and covariance and priors updates and only taking a gradient step for $\mathcal{L}$ every 30 timepoints, we are able to run at close to 1 kHz (NPb). All other bars show example timings from Van der Pol synthetic datasets optimized for speed: $10^4$ dim, where we randomly project down to 200 dimensions and used proSVD to project to 10 dimensions for subsequent Bubblewrap modeling learning; $N = $ 20k, 10k, and 1k nodes, showing how our algorithm scales with the number of tiles; and Prediction, showing the time cost to predict one step ahead for the $N = $ 1k case.

Kalman filtering, (e.g., [16]), and these methods are known to work well in many practical cases, they also imply strong assumptions about the evolution of activity within these systems. Thus, many studies that employ less constrained behavior or study neural activity with less robust dynamics may benefit from more flexible models that can be trained while the experiment is running.

Here, to address this need, we have introduced both a new dimension reduction method that rapidly produces stable estimates of features and a method for rapidly mapping and charting transitions on neural manifolds. Rather than focus on moment-by-moment prediction, we focus on estimating a coarse tiling and probability flow among these tiles. Thus, Bubblewrap may be less accurate than methods based on dynamical systems when state trajectories are accurately described by smooth vector fields with Gaussian noise. Conversely, when noise dominates, is multimodal, or only large-scale probability flow is discernible over longer timescales, Bubblewrap is better poised to capture these features. We saw this in our experiments, where the model of [30] exhibited better overall performance in the mouse video dataset (Figure 3c) when it did not learn to predict and degenerated to a random walk. Indeed, the most relevant comparison to the two approaches is the duality between stochastic differential equations and Fokker-Planck equations, where ours is a (softly) discretized analog of the latter. Nonetheless, in many of the cases we consider, Bubblewrap produces superior results even for state prediction. Nonetheless, like many similar models, ours includes multiple hyperparameters that require setting. While we did not experience catastrophic failure or sensitive dependence on parameters in our testing, and while our methods adapt to the scale and drift of the data, some tuning was required in practice.

As detailed above, while many methods target population dynamics, and a few target closed-loop settings [31, 16, 55], very few models are capable of being trained online. Thus, the most closely related approaches are those in [30, 32], to which we provide extensive comparisons. However, these comparisons are somewhat strained by the fact that we provided all models with the same proSVD-reduced low-dimensional data, while [32] is capable of modeling high-dimensional data in its own right and [30] was targeted at inferring neural computations from dynamical systems. We thus view this work as complementary to the dynamical systems approach, one that may be preferred when small distinctions among population dynamics are less important than characterizing highly noisy, non-repeating neural behavior.

Finally, we showed that online training of Bubblewrap can be performed fast enough for even kiloHertz data acquisition rates if small latencies are tolerable and gradient steps can be performed for small numbers of samples at a time. Yet, for real-time applications, it is not training time but the

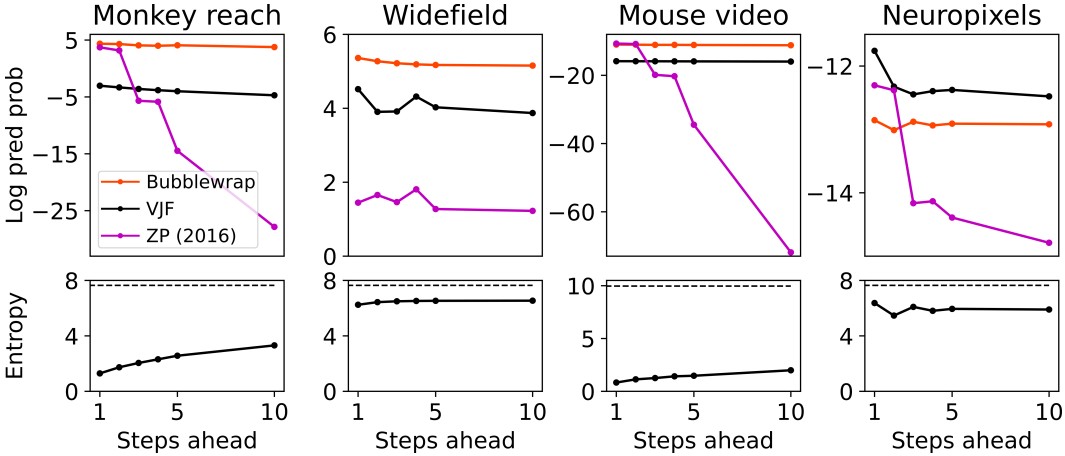

Figure 5: **Multi-step ahead predictive performance. (top)** Mean log predictive probability as a function of the number of steps ahead used for prediction for each of the four experimental datasets studied. Colors indicate model. **(bottom)** Bubblewrap entropy as a function of the number of steps ahead used for prediction. Higher entropy indicates more uncertainty about future states. Dashed lines denote maximum entropy for each dataset (log of the number of tiles).

time required to make predictions that is relevant, and we demonstrate prediction times of tens of microseconds. Moreover, Bubblewrap is capable of producing effective predictions multiple time steps into the future, providing ample lead time for closed-loop interventions. Thus, coarse-graining methods like ours open the door to online manipulation and steering of neural systems.

## Acknowledgments and Disclosure of Funding

Research reported in this publication was supported by a NIH BRAIN Initiative Planning Grant (R34NS116738; JP), and a Swartz Foundation Postdoctoral Fellowship for Theory in Neuroscience (AD). AD also holds a Career Award at the Scientific Interface from the Burroughs Wellcome Fund.

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
