# A Details of proSVD algorithm

We follow the notation of [20, 22] with the exception that we use $X$ for the data matrix rather than $A$, with new data $X_+$ of dimension $n \times b$ rather than $m \times l$. Here, we focus on the updates of the left singular subspace spanned. Details for the right singular subspace are similar and covered in the original works. Matrix sizes are listed for convenience in Table 2.

Table 2: Matrix dimensions for incremental SVD.

| matrix | rows | columns |
|---|---|---|
| $X$ | $n$ | $T$ |
| $X_0$ | $n$ | $k$ |
| $X_+$ | $n$ | $b$ |
| $C$ | $k$ | $b$ |
| $Q_t$ | $n$ | $k$ |
| $R_t$ | $k$ | $k$ |
| $\hat{Q}$ | $n$ | $k+b$ |
| $\hat{R}$ | $k+b$ | $k+b$ |
| $U$ | $k+b$ | $k+b$ |
| $U_1$ | $k+b$ | $k$ |
| $U_2$ | $k+b$ | $b$ |
| $T_u$ | $k$ | $k$ |
| $S_u$ | $b$ | $b$ |
| $G_u$ | $k+b$ | $k+b$ |
| $G_{u_1}$ | $k+b$ | $k$ |

The goal of the algorithm is to maintain an approximation of the data up to the present moment as

$$X \approx QRW^\top \tag{8}$$

with $Q$ and $W$ orthogonal but $R$ not necessarily diagonal. The requirement that (8) be equivalent to the top-$k$ SVD requires that the top-$k$ SVD of $X$ can easily be computed via the SVD of the small matrix $R = U\Sigma V^\top$.

The algorithm begins with an initial data matrix $X_0$, which is factorized via the QR decomposition

$$X_0 = [Q_0 \quad Q_\perp] \begin{bmatrix} R_0 \\ 0 \end{bmatrix} \mathbb{1}, \tag{9}$$

which has the form (8) if we identify $W_0 = \mathbb{1}$. Thus $Q_0$ forms the initial candidate for a basis for the top-$k$ subspace. On subsequent iterations, the procedure is as follows:

1. Observe a new $n \times b$ data matrix $X_+$.

2. Perform a Gram-Schmidt Orthogonalization of this new data, obtaining $C$, a projection into the previous basis $Q_{t-1}$ and a remainder, $X_\perp$.

3. Perform a QR decomposition $X_\perp = Q_\perp R_\perp$. This gives rise to a new factorization

$$X_t = [X_{t-1} \quad X_+] = \hat{Q}\hat{R}\hat{W}^T \tag{10}$$

   with

$$\hat{Q} \equiv [Q_{t-1} \quad Q_\perp] \tag{11}$$

$$\hat{R} \equiv \begin{bmatrix} R_{t-1} & C \\ 0 & R_\perp \end{bmatrix}. \tag{12}$$

4. From this new factorization, the goal is to block diagonalize $\hat{R}$ and truncate to the upper left block, which is the new top-$k$ singular subspace. This is done by first performing an SVD, $\hat{R} = U\Sigma V^\top$.

5. To allow for old and stale data to decay in influence over time and to prevent unbounded accumulation of variance in $\Sigma$, we follow [40] in multiplying $\Sigma$ by a discount parameter $\alpha$ at each step.

6. As in [20, 22], the goal is to find a matrix $G_u$ (and a counterpart on the right, $G_v$) satisfying

$$G_u^\top U = \begin{bmatrix} T_u & 0 \\ 0 & S_u \end{bmatrix} \tag{13}$$

or the equivalent condition

$$G_u^\top U_1 = \begin{bmatrix} T_u \\ 0 \end{bmatrix} \tag{14}$$

with $U_1$ the first $k$ columns of $U$, which would yield

$$G_u^\top \hat{R} G_v = \begin{bmatrix} T_u \Sigma_1 T_v^\top & 0 \\ 0 & S_u \Sigma_2 S_v^\top \end{bmatrix}. \tag{15}$$

7. This done, the matrices can once again be truncated back to their top-$k$ versions:

$$R_t = T_u \Sigma_1 T_v^\top \tag{16}$$
$$Q_t = \hat{Q} G_{u_1} = \hat{Q} U_1 T_u^\top. \tag{17}$$

What is most important to note in this is that the solution to (14) is not unique. Many choices of $G_u$ (equivalently $T_u$) are possible. In [20, 22], a particular solution to this equation is chosen for computational efficiency. By contrast, proSVD seeks to solve

$$\min_{T_u} \|Q_t - Q_{t-1}\|_F = \min_{T_u} \|\hat{Q} U_1 T_u^\top - Q_{t-1}\|_F, \tag{18}$$

with $\|\cdot\|_F$ denoting the Frobenius norm. As previously stated, this is an Orthogonal Procrustes problem with solution $T = \tilde{U} \tilde{V}^\top$, where $\tilde{U}$ and $\tilde{V}$ are defined by the SVD [38]:

$$\tilde{U} \tilde{\Sigma} \tilde{V}^\top = Q_{t-1}^\top \hat{Q} U_1 = [\mathbb{1}_{k \times k} \quad \mathbf{0}_{k \times b}] U_1. \tag{19}$$

That is, $T$ can be found by performing the SVD of the upper left $k \times k$ block of $U$, and the only additional cost of our formulation relative to [20, 22] is the $\mathcal{O}(k^3)$ cost of this SVD, which is negligible in practice for $k$ small (Figure 1b).

## B   Details of Bubblewrap initialization and heuristics

Our implementation of Bubblewrap neither normalizes nor assumes a scale for incoming data. Thus, in order to choose priors that adjust to the scale of the data, we adopt an empirical Bayes approach. That is, we adjust some parameters of the Normal-Inverse-Wishart priors for each bubble, $\mu_{0j}$ and $\Psi_j$, based on data. More specifically, we first calculate empirical estimates $\bar{\mu}$ and $\overline{\Sigma}$ of the data mean and covariance online, as in line 11 of Algorithm 2. We then update the priors as follows:

**Random walk dynamics on $\mu_{0j}$:** If $\mu_{0j}$ is fixed and identical across all bubbles, then points passing through this concentration of Gaussians will be assigned equally to all, which creates problems for tiling. Here, the idea is to allow the $\mu_{0j}$ to randomly walk, thereby breaking degeneracy among nodes with no data points. However, in order to bound the random walk, we include a decay back toward the current center of mass. More specifically, each of the $\mu_{0j}$ is governed by a biased random walk:

$$\mu_{0j}(t) = (1 - \lambda)\mu_{0j}(t - 1) + \lambda \bar{\mu}(t - 1) + \epsilon_{jt} \tag{20}$$

with $\lambda \in [0, 1]$ and $\epsilon_{jt} \sim \mathcal{N}(0, \eta^2)$, $\mathbb{E}[\epsilon_{jt}\epsilon_{j't'}] = \eta^2 \delta_{jj'}\delta_{tt'}$. Here, $\eta$ is a step size for the random walk that may differ by dimension. Note that, when $\bar{\mu}$ is fixed, the distribution of $\mu_{0j}$ converges to $\mathcal{N}(\bar{\mu}, \mathrm{diag}(\eta^2)/\lambda)$.

In practice, we chose $\lambda = 0.02$ and $\eta = \sqrt{\lambda \, \mathrm{diag}(\overline{\Sigma})}$, so that the diffusion range of bubbles with no data is set by the scale of the data distribution.

**Bubble packing for $\Psi_j$:** Here, the basic idea is to choose $\Psi_j$, which controls the covariance around which the prior concentrates, so that the total volume of bubbles scales as the volume of the data. Specifically, let $\Psi_j = \sigma^2 \mathbb{1}$, so that the covariance prior is spherical with radius $\sigma$. We also assume

that, in $k$ dimensions, data take up $\mathcal{O}(L)$ space along each dimension, so that $\mathrm{vol}(data) \sim L^k$. Then the volume of each bubble is $\sim \sigma^k$ and the total volume of all bubbles is

$$N\sigma^k \sim L^k \quad \Rightarrow \quad \sigma \sim \frac{L}{N^{\frac{1}{k}}} \tag{21}$$

which yields

$$\Psi_j = \frac{L^2}{N^{\frac{2}{k}}} \mathbb{1}. \tag{22}$$

More generally, we want $\Psi_j$ to reflect the estimated data covariance $\overline{\Sigma}$, so we set

$$\Psi_j = \frac{1}{N^{\frac{2}{k}}} \overline{\Sigma}. \tag{23}$$

## C  Predictive distribution calculations

Both [30] and [32] use a nonlinear dynamical system model parameterized as

$$\mathbf{x}_{t+1} = \mathbf{x}_t + f(\mathbf{x}_t) + \mathbf{B}_t(\mathbf{x}_t)\mathbf{u}_t + \epsilon_{t+1} \tag{24}$$
$$\mathbf{y}_t \sim P(g(\mathbf{C}\mathbf{x}_t + \mathbf{b})) \tag{25}$$
$$f(\mathbf{x}) = \mathbf{W}\phi(\mathbf{x}) - e^{-\tau^2}\mathbf{x}_t \tag{26}$$
$$\mathbf{x}_0, \epsilon_t \sim \mathcal{N}(\mathbf{0}, \sigma^2\mathbb{1}), \tag{27}$$

with $\phi$ a set of basis functions and $g$ a link function. In [32], $\mathbf{B}$ is assumed constant and $\tau \to \infty$. More importantly, the filtered probability $p(\mathbf{x}_t|\mathbf{y}_{\leq t})$ is approximated by a posterior $q(\mathbf{x}_t) = \mathcal{N}(\boldsymbol{\mu}_t, \mathrm{diag}(\mathbf{s}_t))$ with $\boldsymbol{\mu}_t$ and $\mathbf{s}_t$ defined by the output of a neural network trained to optimize a variational lower bound on the log evidence.

In [30], the loss function is mean squared error in the prediction (24), which is equivalent to the likelihood model

$$p(\mathbf{x}_{t+1}|\mathbf{x}_t) = \mathcal{N}(\mathbf{x}_t + f(\mathbf{x}_t) + \mathbf{B}_t(\mathbf{x}_t)\mathbf{u}_t, \sigma^2\mathbb{1}), \tag{28}$$

with $\sigma^2 = \mathrm{var}[\epsilon_t]$ estimated from the data. In [30], there is no separate observation model (25), so $\mathbf{y} = \mathbf{x}$, and we use (28) (with $\sigma^2$ estimated from the residuals of (24) via an exponential smooth) to calculate predictive likelihood for our comparisons.

For [32], following the procedure outlined there, we can compute the log predictive probability via sampling

$$\mathbf{x}_t^s \sim q(\mathbf{x}_t) \tag{29}$$
$$\mathbf{x}_{t+1}^s \sim \mathcal{N}(\mathbf{x}_t^s + f(\mathbf{x}_t^s) + \mathbf{B}_t\mathbf{u}_t, \sigma^2) \tag{30}$$
$$\log p(\mathbf{y}_{t+1}|\mathbf{y}_{\leq t}) \approx \log\left[\frac{1}{S}\sum_{s=1}^{S} P(\mathbf{y}_{t+1}; g(\mathbf{C}\mathbf{x}_{t+1}^s + \mathbf{b}))\right] \tag{31}$$

with $S = 100$ samples. However, for direct comparison with our method, which operates in a reduced-dimensional space, we compared the predictive probability by feeding all methods the same data as reduced by proSVD, so that $\dim(\mathbf{y}) = \dim(\mathbf{x}) = \dim(data)$.

When predicting more than one time step ahead, we use sequential sampling for both models. For the model of [30], we iterate (24) to produce $S$ trajectories $T-1$ time steps into the future (ending at $x_{t+T-1}^s$ and calculate

$$p(\mathbf{x}_{t+T}|\mathbf{x}_{\leq t}) = \frac{1}{S}\sum_{s=1}^{S} p(\mathbf{x}_{t+T}|\mathbf{x}_{t+T-1}^s) = \frac{1}{S}\sum_{s=1}^{S} \mathcal{N}(\mathbf{x}_{t+T-1}^s + f(\mathbf{x}_{t+T-1}^s), \sigma^2\mathbb{1}) \tag{32}$$

using $S = 100$ trajectories. For the model of [32], we have

$$
\begin{aligned}
p(\mathbf{y}_{t+T}|\mathbf{y}_{\leq t}) &= \int \prod_{t'=t+1}^{T} d\mathbf{x}_{t'} \; p(\mathbf{y}_{t+T}|\mathbf{x}_{t+1:T})p(\mathbf{x}_{t+1:T}|\mathbf{x}_t)p(\mathbf{x}_t|\mathbf{y}_{\leq t}) \\
&\approx \int \prod_{t'=t+1}^{T} d\mathbf{x}_{t'} \; p(\mathbf{y}_{t+T}|\mathbf{x}_{t+1:T})p(\mathbf{x}_{t+1:T}|\mathbf{x}_t)q(\mathbf{x}_t) \\
&\approx \frac{1}{S} \sum_{s=1:S} p(\mathbf{y}_{t+T}|\mathbf{x}_{t+T}^s) \\
&= \frac{1}{S} \sum_{s=1}^{S} P(\mathbf{y}_{t+T}; g(\mathbf{C}\mathbf{x}_{t+T}^s + \mathbf{b})),
\end{aligned} \tag{33}
$$

where again we have used sampling to marginalize over the intervening latent states.

## D    Additional Experiments

All experimental simulations were run on custom-built desktop machines running either Ubuntu 18.04.4 LT or Ubuntu 20.04.2. The computers ran with either 64GB or 128 GB of system memory, and various CPUs including a 4 or 8 core 4.0 GHz Intel i7-6700K processor and 14 core 3.1 GHz Intel i9-7940X processors. GPUs used included an NVIDIA GeForce GTX 1080 Ti (11 GB), 2x NVIDIA RTX 2080 Ti (11 GB), an NVIDIA Titan Xp (12 GB), and an NVIDIA RTX 3090 (24 GB).

### D.1    Datasets

The monkey reach dataset can be found at `https://churchland.zuckermaninstitute.columbia.edu/content/code` [48], and was originally presented in [49]. The widefield calcium imaging and mouse video datasets can both be found at `https://dx.doi.org/10.14224/1.38599` [51], and were originally presented in [50]. The neuropixels dataset can be found at `https://doi.org/10.25378/janelia.7739750.v4` [52], and has a (CC BY-NC 4.0) license. It was originally presented in [53].

To motivate the number of dimensions of the low-dimensional subspace onto which we project the data, we show the cumulative variance explained by each eigenvector of the real datasets in Figure S1. These spectra show that keeping on the order of $\sim 10$ components retains $40\%$ of variance of the monkey reach and neuropixels datasets, and retains $> 99\%$ variance of the widefield and mouse video datasets.

### D.2    proSVD stability

Here we apply proSVD and the incremental block update method of [20, 22] (referred to as streaming SVD) to the experimental datasets used in the main text. The first column of Figure S2 shows that the proSVD basis vectors stabilize after a few tens of seconds for these datasets, with only small gradients thereafter. The second column shows that proSVD basis vectors stabilize to their final (offline) positions faster than streaming SVD vectors.

### D.3    Log predictive probability & entropy

Here we include results for log predictive probability and entropy over time for the datasets not included in the main text. Figure S3 shows the log predictive probability across all time points for all datasets except the 2D Van der Pol and 3D Lorenz (5 % noise) datasets (cp. Figure 2). Figure S4 plots the entropy over time for the 2D Van der Pol (0.05, 0.20) and the 3D Lorenz (0.05, 0.20) datasets ; all other entropy results were shown in the main text (Figure 3d-f, Figure 4b).

### D.4    Final transition matrices

Figure S5 shows the eigenspectra of the final learned transition matrices $A$ for each of the four experimental datasets studied in the main text. The monkey reach and mouse video results yielded

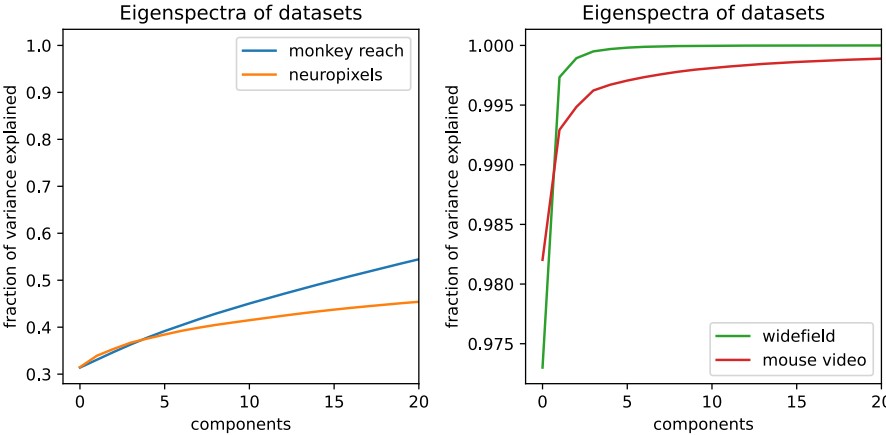

Figure S1: **Eigenspectra of datasets.** Cumulative variance explained by retaining differing numbers of components in linear dimensionality reduction. Labels reference datasets as in the main text.

slowly-decaying eigenspectra, suggesting long-range predictive power, whereas the widefield and neuropixels eigenspectra decayed much faster (and thus yielded less long-range predictive power over time). Figures 3 and 4 in the main text also show this distinction: the monkey reach and mouse video entropy results reached much lower values than the other two datasets, corresponding to lower-uncertainty predictions.

Additionally, we visualize the final node locations and transition entries in $A$ for each experimental dataset in Figure S6. Note that while all timepoints of the data are plotted (grey), only the end result of Bubblewrap is shown (black dots, blue lines).

### D.5 Degeneration of dynamical systems to random walks

For the Lorenz and Mouse video datasets, the model of [30] outperformed both Bubblewrap and VJF in log predictive probability. However, as shown in Figure S7, this is because the predicted step size ($f(x)$ in (24)) drops to near 0. That is, the model degenerates to a random walk. Datasets where this occurs are marked with a * in Table 1.

### D.6 Model benchmarking

Here we present average runtimes for the comparison models on selected datasets, similar to Figure 4c in the main text. Note that we gave all models the already dimension-reduced data. Figure S8 shows only the time needed to update the model for each new datapoint of the 2D Van der Pol (VdP) oscillator dataset for two different numbers of basis functions (labeled '500' and '50'), which are roughly equivalent to the notion of nodes in Bubblewrap. Next, we include the time to generate predictions via sampling, increasing the total computation time, for the Neuropixels dataset ('NP') and the Van der Pol oscillator (0.05 noise), again with 500 or 50 basis functions ('VdP 500' and 'VdP 50'). Finally, we plot the average time to fit Bubblewrap (including prediction time) for the base VdP ('VdP') case using 1k nodes. All comparison model cases, using our implementations, showed that models ran at rates slower than 1 kiloHerz.

### D.7 Bubblewrap model evolution

As shown in the above log predictive probability plots, Bubblewrap takes only a small amount of data to successfully fit the underlying neural manifold. Figure S9 shows our model fit as a function of the number of timepoints seen for the 2D Van der Pol and 3D Lorenz simulated datasets (0.05), with tiling locations settling in after observing a few hundred data points and uncertainty being refined throughout the course of the dataset.

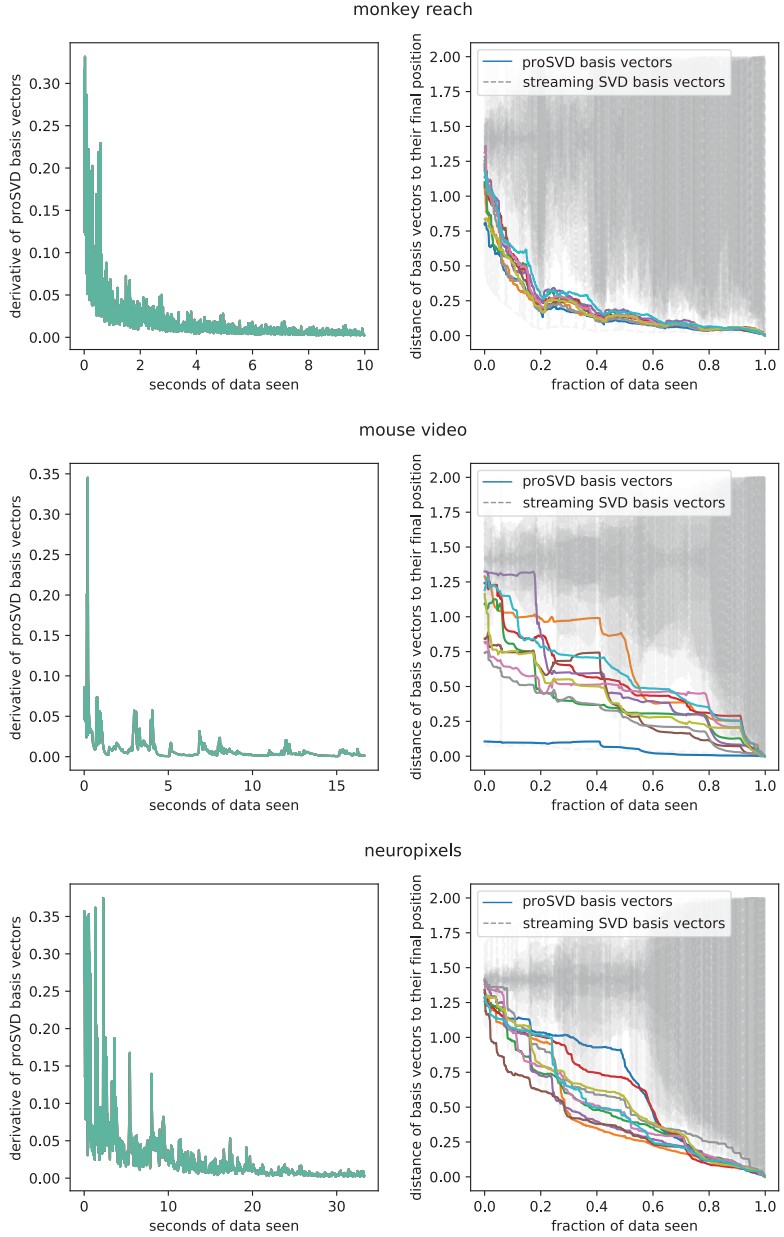

Figure S2: **Rapid stability of proSVD.** Titles reference data as in the main text. First column shows the derivatives of the proSVD basis vectors over time, i.e., the norms of the first difference of the basis vectors learned over time. Derivatives of streaming SVD vectors not shown for clarity. Second column shows how close the streaming vectors are to their final positions, measured by the Euclidean distance from the whole data basis vectors to the most recently learned basis vectors. $k = 10$ vectors shown.

## D.8 Different seeds

To examine the consistency in the performance of Bubblewrap and the two comparison models, we generated a set of 100 trajectories per dataset using different random seeds to set initial conditions. Figure S10 shows the mean log predictive probability for each of these trajectories for the 2D Van der Pol and 3D Lorenz (0.05) datasets.

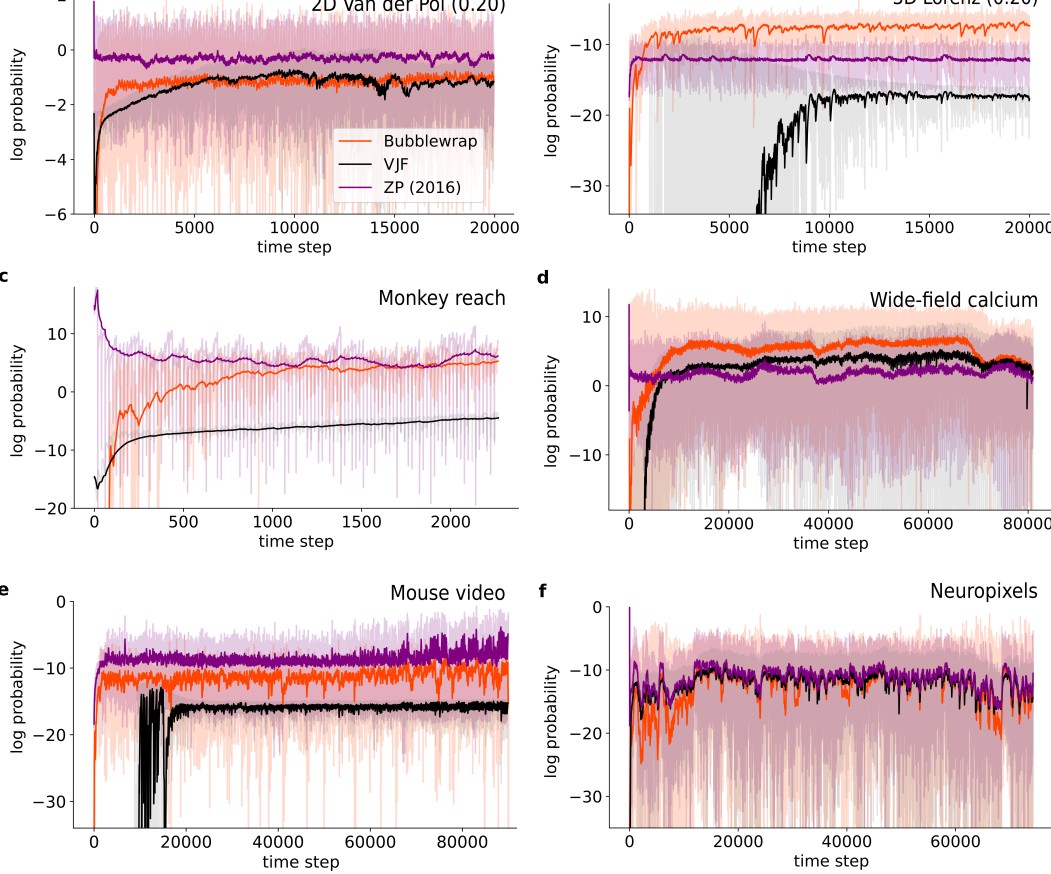

Figure S3: **Log predictive probability results** for all systems not shown in the main text. Log predictive probability across all timepoints for **a)** 2D Van der Pol (0.20 noise), **b)** 3D Lorenz (0.20 noise), **c)** Monkey reach, **d)** Wide-field calcium, **e)** Mouse video, and **f)** Neuropixels datasets. Conventions are as in Figure 2.

## D.9 Benchmarking

Figure S11 displays a typical breakdown of timing for three separate steps within the Bubblewrap algorithm for single step (a) and batch mode (b). For single step updates, the algorithm observes a new data point, updates the data mean and covariance estimates, and uses these to update the node priors ('Update priors'). The algorithm then performs forward filtering and computing sufficient statistics ('E step'). Finally, the M step involves computing and applying gradients of $\mathcal{L}$. ('$\mathcal{L}$ gradient'). In batch mode, prior updates and gradient calculations are performed only once per batch, amortizing the cost of these updates. Finally, we display the time for calculating the model's predicted probability distribution over the tile index at the next time step $p(z_{t+1}|x_{1:t})$ ('Prediction'). This cost is on the order of microseconds.

## D.10 Bubblewrap without heuristics

As noted in the main text, we employed heuristics in addition to the online EM updates to improve Bubblewrap's performance. Figure S12 shows results on the 2D Van der Pol (0.05) dataset with some features removed; 'Bubblewrap' in (a) is plotted for comparison with all features enabled.

When tiles are initialized, their priors place them at the center of mass of the earliest few data points. With each new data point, we re-estimate the data mean and covariance and use these to update the priors for $\mu_j$ and $\Sigma_j$. This includes random walk dynamics on $\mu_{0j}$ (cf. (20)) to break degeneracy among tiles that have seen the same or no data. If we remove this feature, we are still able to

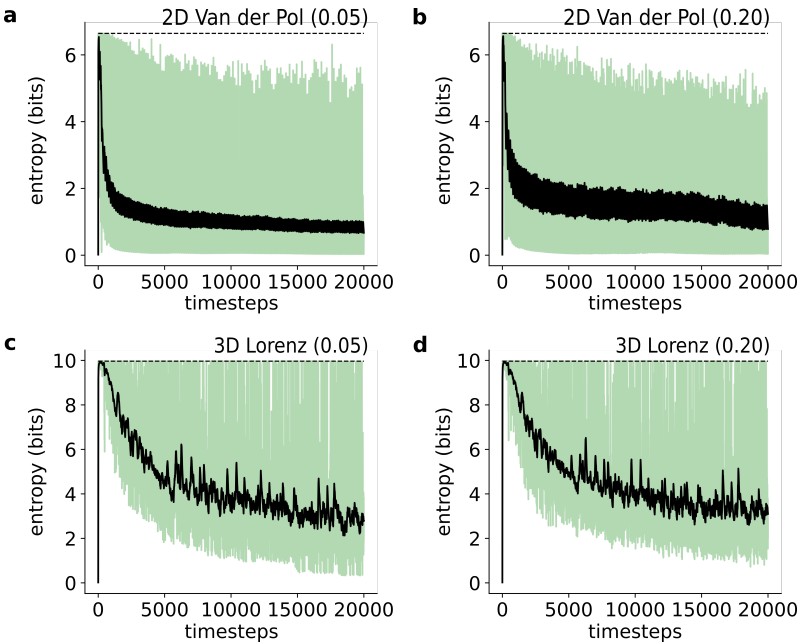

Figure S4: **Entropy** across all timepoints for each comparative model for all systems not shown in main text. Black lines are exponential weighted moving averages of the data. **a)** 2D Van der Pol (0.05), **b)** 2D Van der Pol (0.20), **c)** 3D Lorenz (0.05), **d)** and 3D Lorenz (0.20) datasets.

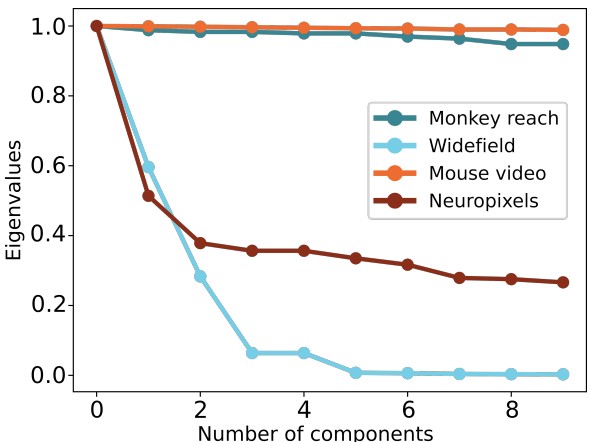

Figure S5: **Eigenspectra of final learned transition matrices.** Labels reference datasets as in the main text. Both the mouse video and monkey reach datasets have many eigenvalues close to 1, implying slow mixing of the corresponding Markov chain and long-range predictive power. By contrast, the widefield calcium and neuropixels datasets, with faster-decaying eigenspectra, quickly lose predictive information over time.

effectively tile the space ('No update priors', Fig. S12b) but a number of redundant, overlapping nodes are left in their initial state (black arrow). For timing considerations, we note that turning off these prior updates provides a speed benefit without drastically diminishing performance in some cases (Fig. S12e).

A second important heuristic we employed was to begin by marking all nodes as available for teleportation and thereby 'breadcrumb' the initial set of incoming datapoints. With this teleport feature turned off ('No teleporting', Fig. S12c), again Bubblewrap is able to effectively learn the correct tiling, though at a slower rate during initial learning (Fig. S12e). Additionally, because

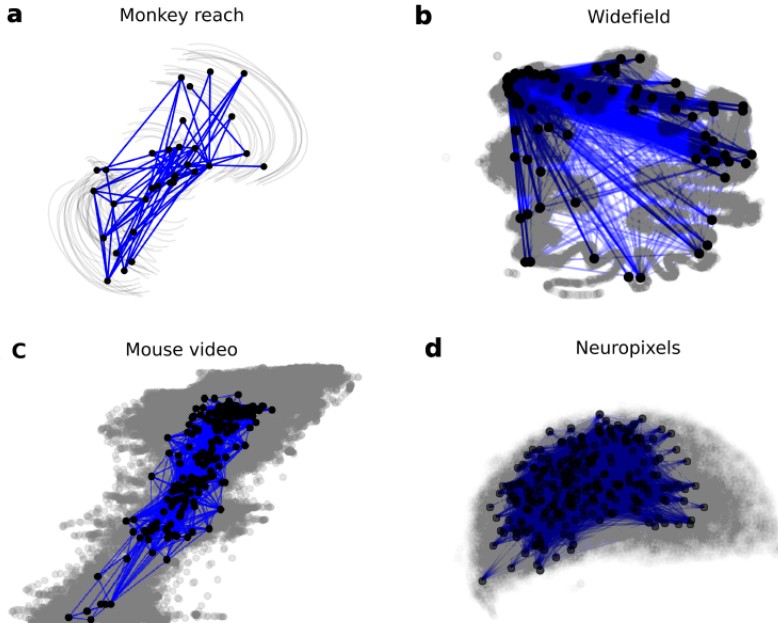

Figure S6: **Final transition matrices for each experimental dataset.** Labels reference datasets as in the main text. Data are plotted in grey as trajectories **(a)** or individual points **(b-d)**. Black dots are the final node locations and blue lines are entries in the final transition matrix A greater than the initialization value $\frac{1}{N}$.

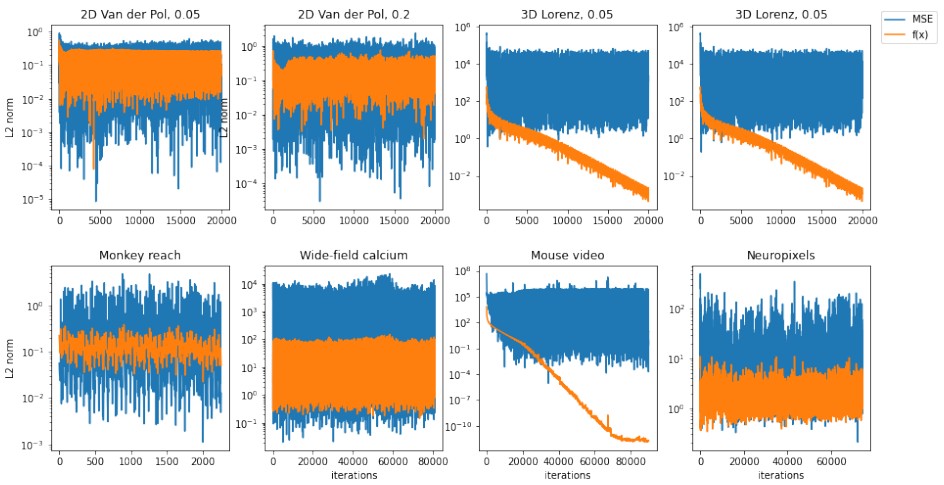

Figure S7: **ZP2016 degenerates to a random walk for some data sets.** Plots of ZP2016 prediction error (MSE, blue curves) and predicted step size ($f(x)$, orange curves, cf. (24)) for **a)** 2D Van der Pol (0.05), **b)** 2D Van der Pol (0.2), **c)** 3D Lorenz (0.05), **d)** 3D Lorenz (0.2), **e)** Monkey reach, **f)** Wide-field calcium, **g)** Mouse video, and **h)** Neuropixels data sets.

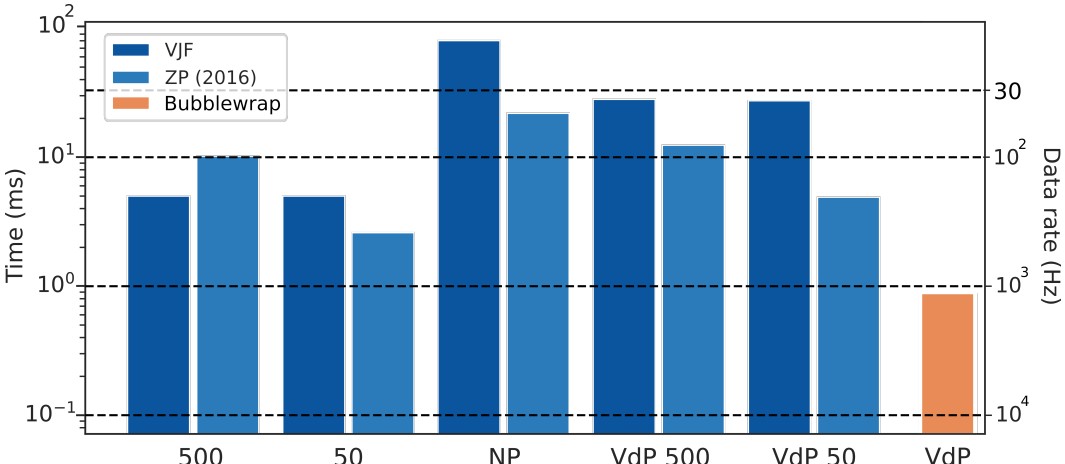

Figure S8: **Benchmarking of comparison models.** Average runtimes for the time to update the model for 500 ('500') or 50 ('50') basis functions on the 2D Van der Pol (0.05 noise) dataset; for the total time including prediction on the Neuropixels dataset ('NP'), the 2D Van der Pol dataset with 500 ('VdP 500') or 50 ('Vdp 50') basis functions; and for Bubblewrap on the base case of the 2D Van der Pol dataset using 1k nodes ('VdP').

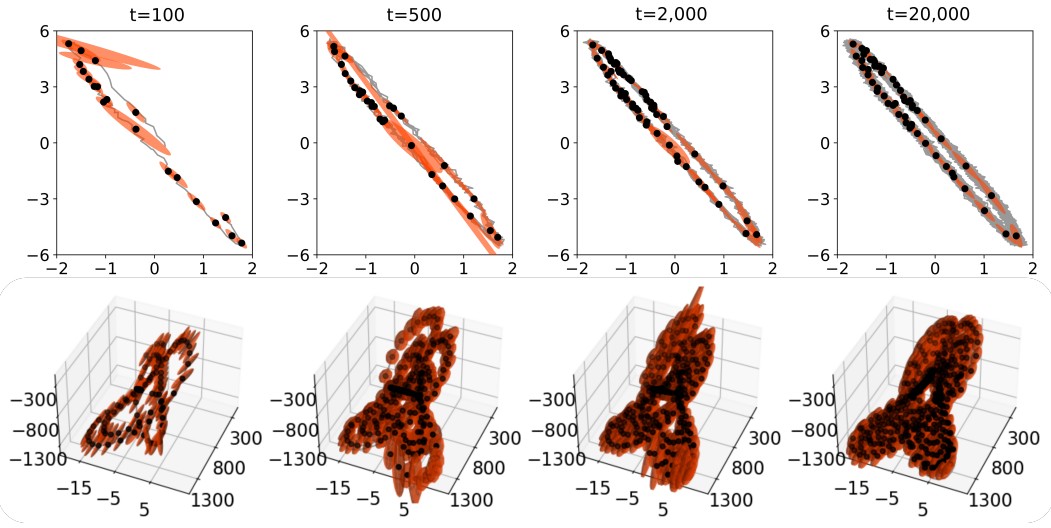

Figure S9: **Bubblewrap over time for simulated datasets.** Model results after t=100, 500, 2000, and 20000 (entire dataset) data points seen. Top row: 2D Van der Pol; bottom row: 3D Lorenz; both 0.05 noise cases.

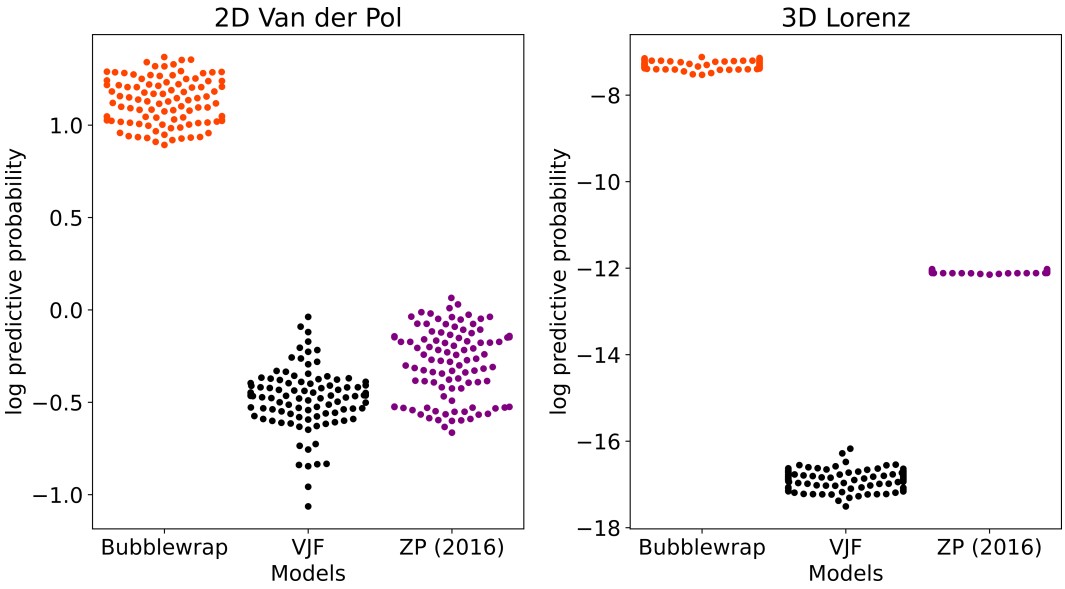

Figure S10: **Mean log predictive probability across seeds.** Model results for 100 different simulated trajectories of the 2D Van der Pol (0.05 noise) and 3D Lorenz (0.05) datasets.

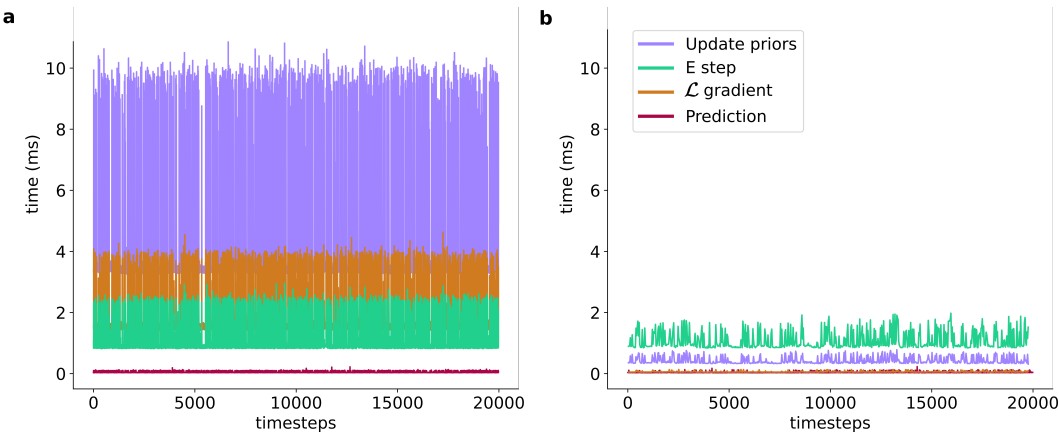

Figure S11: **Timing for different steps inside Bubblewrap. a)** Raw step times for updating priors, E step, gradient update of $\mathcal{L}$, and prediction for the 2D Van der Pol datatset. **b)** Same as in (a), but in batch mode, where prior updates and the gradient step are performed every 30 points.

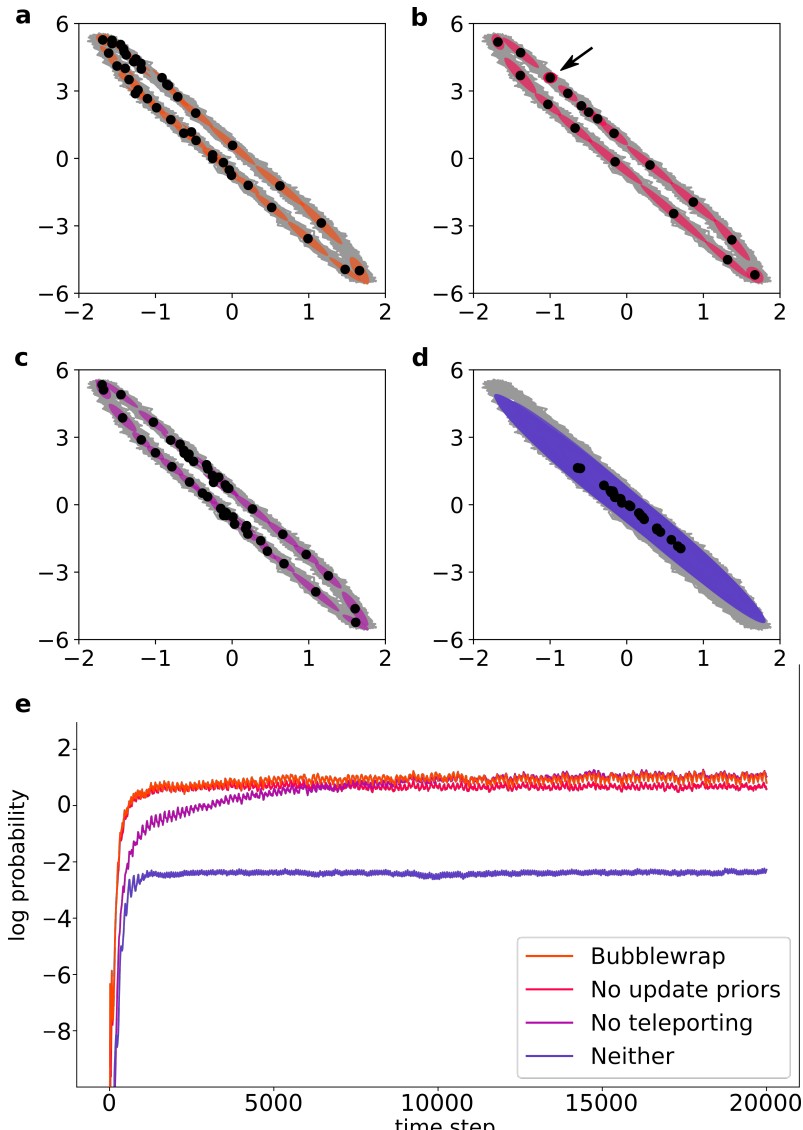

Figure S12: **Bubblewrap feature comparisons.** Model results for the 2D Van der Pol (0.05 noise) data set shown for different combinations of Bubblewrap heuristics. **a)** Reproduction of Figure 2a) with all features enabled. **b)** Results if priors are not updated with data estimates. **c)** Results if teleporting is not employed. **d)** Results if neither prior updates nor teleporting features are enabled. **e)** Smoothed log predictive probability across all time points for the 4 cases shown above (raw data omitted for clarity).

tile locations must be gradually updated through gradient updates on $\mathcal{L}$ rather than instantaneously updated via teleporting, the entire computation time per new data point is roughly 3 times slower.

Finally, if we employ neither of the above features, we can fall into less-optimal tilings with lower log predictive probability results ('Neither', Fig. S12d,e). Here many nodes have all tried to encompass all of the data simultaneously, with heavily overlapping covariance bubbles.