# OpenReview forum: "Bubblewrap: Online tiling and real-time flow prediction on neural manifolds"
_NeurIPS.cc/2021/Conference — NeurIPS 2021 Poster_

### Official Review · Reviewer_oLEL · 2021-07-14

**Rating:** 7
**Confidence:** 4

**Summary:**

The authors present an online method for estimating low-dimensional neural dynamics that scales well to large observation dimensions and high sampling rates. The first step of their method is an online version of SVD, designed so that it converges to a stable subspace with very little training data. The second step of their method is an online version of an HMM with a large number of states that tile the low-dimensional space provided by SVD. They demonstrate their method on both simulated and real data, demonstrating both its speed and accuracy.

**Limitations And Societal Impact:**

The authors have addressed some of the limitations in the discussion. One thing I'm curious about that did not seem to be addressed was the sheer number of states in the HMM - >1k!! In L49 there is mention of a "sparse transition graph", but I found no further mention of this in the main text or the supplemental. It would be useful if the authors could briefly mention how they handle such a large transition matrix, both during training and inference.

**Main Review:**

originality: this work is based on a growing literature of online algorithms designed to capture neural dynamics; the contribution here is a semiparametric model that is not based on a dynamical system (such as LDS), and the authors show their model works better in the high noise regime.

quality: the model is well-motivated and sources methods from a variety of disciplines (random matrix theory, linear algebra, graphical models). The main feature of the model is its online ability to fit data, and with this in mind the authors do a good job of benchmarking the model, as well as comparing it to relevant baselines. However, I am still left wondering what a potential use-case for this method is - perhaps the authors could give an example in the introduction?

clarity: the manuscript is well written and the model is well-motivated. One point of confusion for me was L123-127: the authors mention both a GMM and an HMM, and I first I thought we were dealing with an HMM with GMM emissions; however, it appears that the emissions are standard Gaussians. Maybe the GMM terminology could be dropped in this case?

significance: online estimation of neural dynamics/state is crucial for causal experiments, and as such successful methods will likely be widely adopted. This method seems well suited to such estimation, though the authors have not shown if their method is accurate enough to enable the targeted interventions required for a specific experiment.

**Time Spent Reviewing:**

1.5

---

> ### Author Response · Authors · 2021-08-10
> **Response to Reviewer 3 (oLEL)**
>
> **Summary.** We appreciate the reviewer's comments. Answers to specific questions:
>
> **Use case.** We regret not spelling out a specific use case more clearly. As detailed above in our other responses, our primary use case involves closed loop neuroscience experiments in which one (or potentially many) interventions are triggered based on neural state. For example, holographic photostimulation methods might target multiple neurons, or stimulation might be delivered through many possible combinations of probes. For this purpose, predictive accuracy is important, since this both (a) allows the intervention to be properly timed and (b) potentially increases the time budget for online computations to calculate the optimal intervention. We thus view Bubblewrap as a means of producing a discretized representation that would allow certain optimization or reinforcement learning problems to be solved efficiently online.
>
> **HMM vs. GMM.** What we meant to indicate was that the conditional predictive distribution is a mixture of Gaussians, one per potential successor node. We regret that this phrasing caused confusion for multiple reviewers.
>
> **Number of states.** In practice, we found that, regardless of the number given, the model used only a few hundred to a couple of thousand nodes. Adding additional nodes did not appreciably increase the log predictive probability. However, we found that more complex datasets (e.g. non-trial-aligned, such as the neuropixels dataset) benefited from more nodes (thousands) to tile the space. We simply view this as a consequence of modeling a more complex data density. We thus wanted to ensure that future experiments, which might benefit from more nodes, could also be run efficiently online.
>
> **Transition matrix.** We were surprised that the transition matrix does not require any special handling in our implementation. Even a $1000 \times 1000$ matrix of float32s is only about 4 MB, so in practice we just use the dense representation. However, the transitions themselves are indeed sparse, and were larger graphs necessary, we could use sparse storage formats with minimal changes.
>
> **Experimental efficacy.** The reviewer points out that it is difficult to tell from the data we present whether our predictive accuracy would suffice for the kinds of experiments we have in mind. We agree with this --- every application has different constraints --- but since our initial submission, we have also been able to establish that Bubblewrap performs surprisingly well in predicting future neural activity up to tens of time steps into the future. For example, in the mouse image data set, where the model of [30] outperforms Bubblewrap on one-step-ahead prediction by learning a random walk, the trend reverses when predicting more than four time steps into the future: the log likelihood of Bubblewrap decays from -11.0 at lag 1 to -11.2 at lag 10, while the model of [30] drops from -10.6 to below -12 over the same range. Moreover, entropy in the Bubblewrap transition graph grows from around 0.8 bits at lag 1 to only 2 bits at lag 10 (again, chance is 7.5 bits). In fact, this only rises to 4 bits at lag 100. Thus, Bubblewrap maintains at least some predictive accuracy over hundreds of milliseconds on this challenging data set (with comparable results in our other real data cases). We believe this provides strong evidence that Bubblewrap could be useful in intervening over realistic experimental timescales.

---

### Official Review · Reviewer_KUqF · 2021-07-16

**Rating:** 6
**Confidence:** 4

**Summary:**

The authors present Bubblewrap which is a method for dimensionality reduction and discrete approximation of dynamics for high dimensional timeseries datasets. Bubblewrap stabilizes incremental basis estimates into an incremental subspace estimate and the authors show that this converges quickly in practice. The authors then use an HMM with Gaussian emissions to model the projected data along with several heuristics to achieve better performance. The method in benchmarked in terms of likelihood and latency/data rate on synthetic and real neural and behavioral data.

**Limitations And Societal Impact:**

I would like to see more analysis or discussion about how the choice of discrete inferred states may impact potential uses for this method. Can the sequence of inferred centroids be used? Are the centroids or networks of centroids related to behavior?

**Main Review:**

The manuscript is clearly written and methods that can be run in online during high dimensional neural recordings are important for closed-loop experiments. The reported results are clear and although I have not run the code, it looks complete and uses publicly available datasets.

I'm not an expert in closed-loop neuroscience experiments, but it seems like methods might potentially trade-off
1. latency,
2. the ability to predict the future states of the data Bubblewrap is modeling (e.g., neural states for optogenetic manipulation), or
3. the ability to predict paired timeseries like behavior (e.g., for robotic or task manipulations during neural recordings).

Run-time comparisons for 1. versus refs [30, 32] could be made. 2. is analyzed clearly for this method.

In my opinion, the main area where the manuscript could be strengthened is by including some evaluation of the utility of the inferred discrete states (i.e., 3.). Can the behavioral state of the animal be predicted? The increased log predictive probability compared to other online methods is promising, but some analysis of the ability to predict behavior would make the method more compelling.

General comments/questions
- Lines 126-127: I think many people would consider a HMM with Gaussian emissions to be a state-space model with a noise distribution. Can this sentence be made more precise? The related sentence on lines 214-215 could also be made more clear, I believe the continuous versus discrete dynamics distinction is more salient than moment-to-moment prediction versus tiling. Ultimately, Bubblewrap also learns the tiling through maximizing the data likelihood, which is another way to say prediction. I think the sentence on lines 120-122 has a more clear statement of the comparison.
- Could you provide some interpretation of the plots of entropy in Figs 3 and 4? Is it meaningful that Fig 3 d and f drop closer to zero while Fig 3 e stays closer to the max? Does it tell us something about the neural dynamics?
- In Fig S2, what is the grey in the background of the right column mean?
- Lines 225-226: can you expand on what the most important parameters to tune were, if any?
- The caption for the plots of log predictive probability and entropy don't describe what the black points are? Smoothed average?

Minor comments/questions
- Is there a reason that sparse random projections are used for the initial projection rather than dense random projections? Is there a computational benefit?
- For Fig 4 c, having stacked bars and a log-scale y axis makes it difficult to compare times if they don't happen to be anchored to the same lower point. Having 4 bars (3 parts + total time) grouped side-by-side for each case may help. I would have also expected the "Prediction" time to scale with N, am I misunderstanding something?

**Time Spent Reviewing:**

5

---

> ### Author Response · Authors · 2021-08-10
> **Response to Reviewer 2 (KUqF)**
>
> **Summary.** We appreciate the reviewer's comments. The reviewer is correct that our primary use case is in closed-loop neuroscience experiments. These include, as the reviewer suggests, triggering interventions like optogenetic stimulation, as well as more complex variants where we select interventions from among a large set (see comments to R1 above). Here, as well as above, we attempt to clarify the utility of the representation learned by Bubblewrap.
>
> **Behavioral modeling.** We agree that one of the primary use cases of the latent representation we learn would be behavioral modeling. While we have not pursued that in this work, we note two possible approaches for future work that benefit from learning a discretized, tiled representation:
>
> 1. We can learn a nonparametric predictive model by fitting a separate predictor within each tile. For typical objectives, this is an online convex optimization problem and can be learned efficiently using gradient-based methods. Moreover, by utilizing the posterior distribution over tiles, $\alpha_t$, we can easily interpolate between these values based on the observed data, mitigating the bumpy nature of the density model pointed out by Reviewer 1.
>
> 2. Alternately, we could use the node locations/tile centers as inducing point locations for a sparse variational Gaussian Process (Hensman, de G. Matthews, Ghahramani, 2015). That is, we learn a GP only at the node locations but share statistical strength smoothly across tiles. This model, too, can be learned online using gradient-based updates of a small number of parameters per node.
>
> Both of these methods would allow us to model behaviors that can be viewed as functions of the neural manifold, but the discrete approximation makes both more flexible while remaining computationally tractable.
>
> **Latency times.** These benchmarks are easy to perform and could be included in a camera-ready revision. As stated in the paper, results from [30] are based on our own implementation using JAX, which is fast. Here, we do not expect to see a significant advantage for our method. One caveat to this is that, in order to make comparisons between predictive likelihoods (which scale with dimension) fairer, we have given all models the same dimension-reduced data. When either [30] or [32] are given raw data, we expect some increase in computation time and decrease in performance for both, since they would then be modeling higher-dimensional data. We can easily include this as supplementary information.
>
>
> **Tiling versus discretization.** We take the reviewer's point here. Of course, there are many possible ways to discretize possible neural states, and our conditional mixture modeling/tiling approach is only one. It may be, as the reviewer suggests, that the discretization is the more important factor. Our particular discretization does lead to a transition graph embedded in the low-dimensional space, which allows us to characterize neural data sets by the degree to which neural dynamics are local and smooth (transitions to nearby nodes in the Euclidean sense) versus discontinuous and ''jumpy'' (neighboring nodes that are distant in the low-dimensional subspace). We think this is a potentially interesting secondary application of this method.
>
> **Entropy plots.** Yes! These are the entropies of the predictive distribution, and thus low numbers indicate a high degree of predictability of the neural dynamics (conditioned on modeling assumptions). Viewed another way, relative to a prior of uniform transitions between nodes, the difference between this number and $\log_2 N$ is the amount of predictive information (per time step) Bubblewrap has captured about the data. These numbers, unlike the predictive probabilities, are more comparable across data sets, and we view them as roughly indicative of the difficulty of the prediction problem in each.
>
> **Plotting clarifications.** The black lines in the log predictive probability and entropy plots are smoothed averages (exponential weighted moving averages) of the data plotted in color. We apologize for not having this information in the caption, and will certainly add it in. The gray (dashed lines) in the background of Figure S2 are the true streaming SVD basis vectors, indicating that these basis vectors change directions very rapidly during the course of an experiment compared to ssSVD basis vectors. For Figure 4c, we are happy to modify this data into side-by-side bar plots vs. the stack bar plots for easier time comparisons. We apologize that the prediction time was ambiguous; it does scale with $N$, and in the figure specifically it is shown for $N = 1000$ nodes.
>
> **Tuning parameters.** As discussed in more detail in our response to Reviewer 3 ('Number of states'), we set $N$, the maximal number of nodes, to be between a few hundred and a few thousand depending on the complexity of the problem. For instance, in the 2d Van der Pol case, we only used 100 nodes, whereas the Neuropixels dataset used 1000 nodes.  In practice, the model often used fewer nodes than $N$. We chose $M$, the initial data buffer, to be small in comparison to the length of the dataset (from tens to a few hundred timepoints). We also tuned the step size used with Adam for optimizing $Q$, using values between $10^{-3}$ and $10^{-1}$. These were the most important parameters to tune. Other parameters --- $\nu, \lambda$ --- were set to small values ($10^{-3})$ and unchanged between datasets. Lastly, $\epsilon_t$, our 'forgetting' parameter, was either set to 0 (no forgetting) or a small value ($10^{-3}$). The choice of $\epsilon$ would in practice depend on the experimenter's goals. Some small forgetting for datasets that shift globally over time would be advantageous when designing real-time interventions. We can certainly add a few sentences to the main text to elaborate on our parameter choices.
>
> **Random projections.** The reviewer is correct: sparse random projections are faster to compute, and this speedup was sizable for our highest-dimension input data.

---

> > ### Comment · Reviewer_KUqF · 2021-08-18
> > **Response to authors**
> >
> > Thanks for the response which addresses my comments/questions. I will raise my score from 5 to 6.
> >
> > Regarding "Tiling versus discretization", my point was just that I think you could make the distinction more clear/unambiguous in the text between your tiling method, which uses a kind of semi-parametric, discrete dynamics versus common continuous dynamics modeling approaches.
> >
> > For instance, this sentence (125-127): "As the number of components in the GMM is increased, the model produces an increasingly finer-grained description of dynamics that assumes neither an underlying state space model nor a particular distribution of noise."
> >
> > I think both your approach could arguably fall under a "state space" approach (HMM) with a "noise distribution" (Gaussian emissions), depending on the readers precise definition of these words. My understanding of the contrast you are trying to make is between your semi-parametric discrete dynamics and a continuous type modeling approach like $\dot x = f(x),\ y= g(x) + \epsilon$.

---

> > > ### Author Response · Authors · 2021-08-19
> > > **Response to clarification**
> > >
> > > Thank you. That makes sense, and we are happy to clarify these points.

---

### Official Review · Reviewer_Xhub · 2021-07-16

**Rating:** 6
**Confidence:** 3

**Summary:**

This paper presents a method to efficiently characterize high-dimensional neural dynamics using a soft "tiling" of a lower-dimensionality manifold. Specifically, the method is a combination of two algorithms, (i) "Stable Streaming SVD", a dimensionality reduction method that aims to find the top-k-dimensional subspace instead of the top-k bases; (ii) "BubbleWrap", a Gaussian mixture model of the probability density with Hidden Markov Model transitions, that works on the low-dimensional manifold generated in (i). This is a model-free approach, in the sense that it is not based on a model of the dynamical system, and the authors claim that the flexibility of the method makes it outperform other (model-based) approaches in high-noise regimes. The authors also highlight that the proposed inference is fast, possibly allowing online applications.

**Ethical Concerns:**

None noted.

**Limitations And Societal Impact:**

The authors discussed limitations of their approach, including the decreased accuracy as a tradeoff to gain flexibility.

I do not see any potential negative societal impact of this work.

**Main Review:**

This work addresses an important and interesting problem in the analysis of neural data: a first-pass, low-dimensional characterization of the neural dynamics, especially when the underlying dynamical system is not well understood and/or when it is important to have a rough picture of the dynamics with a limited time budget. With the flexibility of the model formulation and the fast inference, the method has a potential of being a useful tool in the field.

I am enthusiastic about the first part, ssSVD; this could be a very useful method for characterizing low-dimensional dynamics. The idea of finding a stable subspace, instead of the specific basis set, is simple but effective. The demonstration in Fig 1 is also impressive.

I think the "BubbleWrap" algorithm is an interesting take to the problem, with a novel combination of ideas in the context of neural dynamics. However, the presented results are less convincing. In particular, it is not clear what are lessons from the analyses (and what is the potential use case for the method), or how the presented results support the usefulness of the method. My major concerns are below.

Major concerns:

- Log predictive probability is a key quantity in the analysis in the Experiments section, but it is not sufficiently described in the main paper. (It is understandable to keep the details in the supplements, but the main text should be self consistent.) In particular, it is not clear how one should interpret the log predictive probability. What is the message that should be read out of Table 1? Is the absolute value (mean) meaningful, or should one only compare values across different methods within each row? Also in Fig 3d-f and Fig 4b, the values are not really meaningful without having a sense of "how good is good".

- The authors already note in the paper that the proposed method takes flexibility at the cost of reduced accuracy. But in this case, it should be discussed how the (possibly inaccurate) outcome of this method should be used towards a final understanding of the neural data. For example, in Fig 3a, although the highlighted tiles (red) roughly overlap with the trajectory (blue), there are considerable discrepancy, especially for the lower trajectory; if you do any downstream analysis based on this representation, it is inevitable that these analyses will be flawed. If BubbleWrap is meant to be a preliminary analysis, what follow-up analyses could it guide?

- Related to the accuracy issue: the "bubble" structure, due to the Gaussian-mixture nature of the model, introduces an artificial heterogeneity to the probability density. It would be necessary to discuss the effect of these "bubbles" on further analysis based on the resulting probability, and/or ways to remove this artifact by adding a postprocessing step.

- Fig 3b: Text description (L180) says that this dataset was chosen to demonstration a case with rapid transitions between noise clusters. Because the model explicitly infers transition probabilities, it is important that they are shown and discussed in this case, not "omitted for clarity". Speaking of the transition probabilities, it seems that the black lines in Fig 3a do not always correctly capture the trajectory.

Minor comments:

- L107 and Fig 1c: it is not clear how to read out the tradeoff value around n=200 from this figure. In fact, the color-coded points looks like n is transitioning from somewhere around 10 (blue, first point) to around 500 (red, second point) with no intermediate points in between. In general, please indicate the actual values of n in the figure if the value is relevant.


**EDIT**: The author response to my comments, as well as to the comments from the other reviewers, helped clarify my original concerns, which were:

> it is not clear what are lessons from the analyses (and what is the potential use case for the method), or how the presented results support the usefulness of the method.

As for the first point (use case), it was crucial for me to understand that the target use case of this set of methods was close-loop experiments, which I missed in my initial reading of the paper. Now I better appreciate the potential usefulness of the method in real time application.

As for the second point (results), the authors addressed my specific questions in the response. Also importantly, as I was reading the responses to the other reviewers, I realized that I was misunderstanding the "entropy" in the BubbleWrap results as the entropy of the state-space (x) distribution, where it was the entropy of the predictive conditional (z|x). This makes a lot of sense and it allows me to re-evaluate the results in the paper. Clarification on the HMM/GMM terminology was also helpful.

Given this, I am happy to increase my rating from 4 to 6, and now vote for acceptance, under the assumption that the authors will describe these potentially confusing points more clearly in the final version of the manuscript.

**Time Spent Reviewing:**

5

---

> ### Author Response · Authors · 2021-08-10
> **Response to Reviewer 1 (Xhub)**
>
> **Summary.** We appreciate the reviewer's comments. While the reviewer is enthusiastic about our ssSVD dimension reduction algorithm, the reviewer was less clear about use cases for Bubblewrap and the strength of our results.
>
> First, an update on ssSVD: in our original submission, we neglected to notice that line 10 in Algorithm 1 can be further simplified. Since, from line 7, $\hat{Q} = [Q_{t-1} \quad Q_\perp]$, $M$ in line 10 simply becomes the upper $k \times k$ block in $U$. This obviates the potentially expensive matrix product in line 10 and means that the only added computational cost for ssSVD (relative to the algorithms on which it is based) is the $\mathcal{O}(k^3)$ SVD on line 11. This simplification results in additional speedups, which amount to about a $30\%$ decrease in per-sample computation time for reducing $N=1000$ to $k=10$ dimensions. The performance of the algorithm is thus more impressive than reported.
>
> **Essential use case.** This could have been clearer in the paper: The primary set of applications we envision for this work is in performing closed-loop experiments. These go beyond the typical closed-loop control studies we cite in the introduction and include cases in which we might be choosing from a potentially large set of interventions. In these studies, the ability to predict neural activity some distance into the future (either to increase time budgets or to optimally time interventions) is crucial, and thus we focus on accurate predictive modeling. Furthermore, our use of a discrete representation may be particularly useful for performing reinforcement learning or model predictive control, which are often easier in discretized state spaces. As we have shown, we outperform existing models for this predictive purpose on especially challenging benchmark data sets. Moreover, we also dramatically outperform competing methods when predicting multiple steps into the future (see response to Reviewer 3).
>
> **Log predictive probability.** As the reviewer points out, we discuss this in the supplement. This can easily be moved to the main text in a revised version. The reviewer is also correct that values are only directly comparable within rows of Table 1. We opted for predictive probabilities because this is a metric common to both Bubblewrap and the models to which we compare. However, for Bubblewrap itself, we also provide entropy of the predictive distribution, which we show in all cases to be well below chance. That is, Bubblewrap successfully extracts predictive information on data sets where comparable models learn only a random walk.
>
> **Bubble structure.** The reviewer points out that our use of a conditional Gaussian mixture may poorly approximate a smooth density. This is correct, but is endemic to many density estimation models. Of course, as the number of mixture components grows large, there is a well understood theory for convergence to the true distribution (see, e.g., Genovese \& Wasserman, Ann. Statist. 28(4): 1105-1127, 2000). Moreover, for any given data point, part of the forward algorithm is the calculation of $\alpha_j(t)$ (Eq 5), the posterior probability distribution over nodes, which offers a natural method of either smoothing predictions or interpolating between nodes. Thus, the algorithm itself does mitigate some of its own ''bumpiness.'' In fact, we describe possible approaches to modeling behavior that leverage this in response to Reviewer 2 below.
>
> **Figure 3b.** We found that plotting too much made several of these already busy figures difficult to read. We are happy to include complete transition overlays in the supplementary materials for all experiments. The reviewer is correct: Fig. 3a does show that we do not always correctly capture the trajectory. The plotted fluctuations in the log predictive probability show that at some time points we do better at predicting the next point than others. Yet because of natural fluctuations in the data themselves, Bubblewrap does nearly as well as the dynamical systems-based methods, which need to accommodate all trajectories using a single model. It is also important to note that the monkey reach data in Fig 3a result from application of a highly constrained parametric model (online jPCA) prior to modeling, and are thus closest to the best case scenario for dynamical systems methods. Our other real datasets, which are considerably noisier, are more representative, and in those cases, Bubblewrap nonetheless captures 2.5 to 9 bits of predictive information.
>
> **Figure 1c.** We agree that the color-coding for reading out values for $n$ could have been clearer, and we can certainly add labels to the first two points, or include additional intermediary points. The values currently shown for $n$ are $10, 200, 400, 600, \dots$, showing the tradeoff between $10$ and $200$. In general, the choice of $n$ will depend on the application and the need for accuracy vs. low latency.

---

> > ### Comment · Reviewer_Xhub · 2021-08-23
> > **Response to authors - major concerns clarified**
> >
> > Thank you for the response! I read the authors' responses to my comments as well as to the comments from the other reviewers, and it helped clarify my original concerns, which were:
> >
> > > it is not clear what are lessons from the analyses (and what is the potential use case for the method), or how the presented results support the usefulness of the method.
> >
> > As for the first point (use case), it was crucial for me to understand that the target use case of this set of methods was close-loop experiments, which I missed in my initial reading of the paper. Now I better appreciate the potential usefulness of the method in real time application.
> >
> > As for the second point (results), the authors addressed my specific questions in the response. Also importantly, as I was reading the responses to the other reviewers, I realized that I was misunderstanding the "entropy" in the BubbleWrap results as the entropy of the state-space (x) distribution, where it was the entropy of the predictive conditional (z|x). This makes a lot of sense and it allows me to re-evaluate the results in the paper. Clarification on the HMM/GMM terminology was also helpful.
> >
> > Given this, I am happy to increase my rating from 4 to 6, and now vote for acceptance, under the assumption that the authors will describe these potentially confusing points more clearly in the final version of the manuscript.

---

> > > ### Author Response · Authors · 2021-08-25
> > > **Response to clarifications**
> > >
> > > We will certainly incorporate the above clarifications and additional use case details into a final manuscript. Thank you for all the helpful comments.

---

### Author Response · Authors · 2021-08-10
**Introductory response to reviews**

We appreciate thoughtful and thorough reviews by all three reviewers. All reviewers found the problem we address interesting and important, and found solid evidence for our claim to provide state-of-the-art results in prediction over current models. Moreover, at least one reviewer thought our streaming SVD formulation novel and potentially quite useful. However, reviewers had concerns about a lack of clarity in describing both our intended use case and the utility to which the discrete representation learned by Bubblewrap might be put.

In our responses to each reviewer, we clarify that our desired use case is closed-loop neuroscience experiments, where predicting neural activity in advance might allow for state-dependent interventions to be delivered for testing causal hypotheses. Moreover, we outline three new advances to the work since our initial submission that further strengthen our claims and directly address reviewer concerns:

1. Due to a further simplification, our ssSVD runs even faster than reported in the initial manuscript (see Response to Reviewer 1).

2. We describe how we expect the Bubblewrap representation to be useful in flexibly modeling behavior (see response to Reviewer 2).

3. We can now show that Bubblewrap not only meets or exceeds the performance of existing methods when predicting one time step ahead, it better retains this performance across multiple time steps (see Response to Reviewer 3).

We believe that each of these new findings (the last of which would form a final figure in a camera-ready submission) substantially strengthen our claims.

---

### Decision · Program_Chairs · 2021-09-27

**Decision:**

Accept (Poster)

**Comment:**

After discussion, the reviewers all converged on accept or weak accept ratings for this paper, once the stated changes are implemented. The methodological novelty is somewhat limited, but the problem is important, the approach is sensible, and the results are good.